# Posterior Label Smoothing for Node Classification

## Abstract

Soft labels can improve the generalization of a neural network classifier in many
domains, such as image classification. Despite its success, the current literature
has overlooked the efficiency of label smoothing in node classification with graph-
structured data. In this work, we propose a simple yet effective label smoothing for
the transductive node classification task. We design the soft label to encapsulate
the local context of the target node through the neighborhood label distribution. We
apply the smoothing method for seven baseline models to show its effectiveness.
The label smoothing methods improve the classification accuracy in 10 node classi-
fication datasets in most cases. In the following analysis, we find that incorporating
global label statistics in posterior computation is the key to the success of label
smoothing. Further investigation reveals that the soft labels mitigate overfitting
during training, leading to better generalization performance.

## 1 Introduction

Adding a uniform noise to the ground truth labels has shown remarkable success in training neu-
ral networks for various classification tasks, including image classification and natural language
processing [Szegedy et al., 2016a, Vaswani et al., 2017, Müller et al., 2019, Zhang et al., 2021].
Despite its simplicity, label smoothing acts as a regularizer for the output distribution and improves
generalization performance [Pereyra et al., 2017]. More sophisticated soft labeling approaches have
been proposed based on the theoretical analysis of label smoothing [Li et al., 2020, Lienen and
Hüllermeier, 2021]. However, the usefulness of smoothing has been under-explored in the graph
domain, especially for node classification tasks.

In this work, we propose a *simple yet effective* smoothing method for transductive node classification
tasks. Inspired by the previous work suggesting predicting the local context of a node [Hu et al., 2019,
Rong et al., 2020], such as subgraph prediction, helps to learn better representations, we propose
a smoothing method that can potentially reflect the local context of the target node. To encode
the neighborhood information into the node label, we propose to relabel the node with a posterior
distribution of the label given neighborhood labels.

Under the assumption that the neighborhood labels are conditionally independent given the label
of the node to be relabeled, we factorize the likelihood into the product of conditional distributions
between two adjacent nodes. To compute the posterior, we estimate the conditionals and prior from a
graph's global label statistics, making the posterior incorporate the local structure and global label
distributions. Since the posterior obtained in this way does not preserve the ground truth label, we
finally interpolate the posterior with the ground truth label, resulting in a soft label.

The posterior, however, may pose high variance when there are few numbers of neighborhood
nodes. To mitigate the issue with the sparse labels, we further propose iterative pseudo labeling to
re-estimate the likelihood and prior based on the pseudo labels. Specifically, we use the pseudo labels

of validation and test sets to update the likelihood and prior, along with the ground truth labels of the training set.

We apply our smoothing method to seven different baseline neural network models, including MLP and variants of graph neural networks, and test its performance on 10 benchmark node classification datasets. Our empirical study finds that the soft label with iterative pseudo labeling improves the accuracy in 67 out of 70 cases despite its simplicity. We analyze the cases where the soft label decreases the accuracy and reveals characteristics of label distributions with which the soft labeling may not work. Further analysis shows that using local neighborhood structure and global label statistics is the key to its success. Through the loss curve analysis, we find that the soft label prevents over-fitting, leading to a better generalization performance in classification.

## 2 Related work

In this section, we introduce previous studies related to our method. We begin by discussing various node classification methods, followed by an exploration of the application of soft labels in model training.

### 2.1 Node classification

Graph structures are utilized in various ways for node classification tasks. Some studies propose model frameworks based on the assumption of specific graph structures. For example, GCN [Kipf and Welling, 2016], GraphSAGE [Hamilton et al., 2017], and GAT [Veličković et al., 2017] aggregate neighbor node representations based on the homophilic assumption. To address the class-imbalance problem, GraphSMOTE [Zhao et al., 2021], ImGAGN [Qu et al., 2021], and GraphENS [Park et al., 2022] are proposed for homophilic graphs. $H_2$GCN [Zhu et al., 2020] and U-GCN [Jin et al., 2021] aggregate representations of multi-hop neighbor nodes to improve performance on heterophilic graphs. Other studies concentrate on learning graph structure. GPR-GNN [Chien et al., 2020] and CPGNN [Zhu et al., 2021] learn graph structures to determine which nodes to aggregate adaptively. LDS [Franceschi et al., 2019], IDGL [Chen et al., 2020] and DHGR [Bi et al., 2022] take a graph rewiring approach, learning optimized graph structures to refine the given structure. Besides, research such as ChebNet [Defferrard et al., 2016], APPNP [Gasteiger et al., 2018], and BernNet [He et al., 2021] focus on learning appropriate filters from the graph signals.

### 2.2 Classification with soft labels

Hinton et al. [2015] demonstrate that a small student model trained using soft labels generated by the predictions of a large teacher model shows better performance than a model trained using one-hot labels. This approach, known as knowledge distillation (KD), is widely adopted in computer vision [Liu et al., 2019], natural language processing (NLP) [Jiao et al., 2020], and recommendation systems [Tang and Wang, 2018] for compression or performance improvement. In the graph domain, applying KD has been considered an effective method to distill graph structure knowledge to student models. TinyGNN [Yan et al., 2020] highlights that deep GNNs can learn information from further neighbor nodes than shallow GNNs, and it distills local structure knowledge from deep GNNs to shallow GNNs. NOSMOG [Tian et al., 2023] improves the performance of multi-layer perceptrons (MLPs) on graph data by distilling graph structure information from a GNN teacher model.

On the other hand, simpler alternatives to generate soft labels are considered. The label smoothing (LS) [Szegedy et al., 2016a] generates soft labels by adding uniform noise to the labels. The benefits of LS have been widely explored. Müller et al. [2019] show that LS improves model calibration. Lukasik et al. [2020] establish a connection between LS and label-correction techniques, revealing LS can address label noise. LS has been widely adopted in computer vision [Zhang et al., 2021] and NLP [Vaswani et al., 2017] studies, but has received little attention in the graph domain.

## 3 Method

In this section, we describe our approach for label smoothing for the node classification problem and provide a new training strategy that iteratively refines the soft labels via pseudo labels obtained from the training procedure.

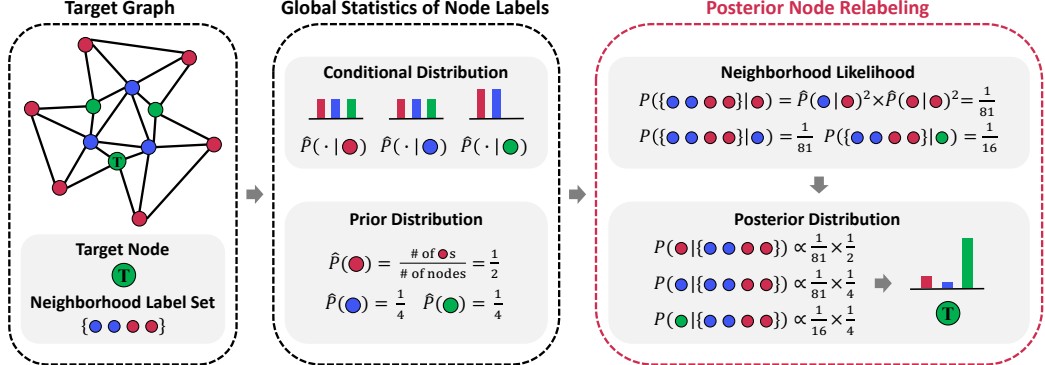

Figure 1: Overall illustration of posterior node relabeling. To relabel the node label, we compute the posterior distribution of the label given neighborhood labels. Note that the node features are not considered in the relabeling process.

## 3.1 Posterior label smoothing

Consider a transductive node classification with graph $\mathcal{G} = (\mathcal{V}, \mathcal{E}, \boldsymbol{X})$, where $\mathcal{V}$ and $\mathcal{E}$ denotes the set of nodes and edges respectively, and $\boldsymbol{X} \in \mathbb{R}^{|\mathcal{V}| \times d}$ denotes $d$-dimensional node feature matrix. For each node $i$ in a training set, we have a label $y_i \in [K]$, where $K$ is the total number of classes. We use the notation $\boldsymbol{e}_i \in \{0, 1\}^K$ for one-hot encoding of $y_i$, i.e., $e_{ik} = 1$ if $y_i = k$ and $\sum_k e_{ik} = 1$. In a transductive setting, we observe the connectivity between all nodes, including the test nodes, without having true labels of the test nodes.

We propose a simple and effective relabeling method to allocate a new label of a node based on the label distribution of the neighborhood nodes. Specifically, we consider the posterior distribution of node labels given their neighbors. Let $\mathcal{N}(i)$ be a set of neighborhood nodes of node $i$. If we assume the distribution of node labels depends on the graph connectivity, then the posterior probability of node $i$'s label, given its neighborhood labels, is

$$P(Y_i = k | \{Y_j = y_j\}_{j \in \mathcal{N}(i)}) = \frac{P(\{Y_j = y_j\}_{j \in \mathcal{N}(i)} | Y_i = k) P(Y_i = k)}{\sum_{\ell=1}^{K} P(\{Y_j = y_j\}_{j \in \mathcal{N}(i)} | Y_i = \ell) P(Y_i = \ell)} \ . \tag{1}$$

The likelihood measures the joint probability of the neighborhood labels given the label of node $i$. To obtain the likelihood, we approximate the likelihood through the product of empirical conditional label distribution between adjacent nodes, i.e., $P(\{Y_j = y_j\}_{j \in \mathcal{N}(i)} | Y_i = k) \approx \prod_{j \in \mathcal{N}(i)} P(Y_j = y_j | Y_i = k, (i, j) \in \mathcal{E})$, where $P(Y_j = y_j | Y_i = k, (i, j) \in \mathcal{E})$ is the conditional of between adjacent nodes. The conditional between adjacent nodes $i$ and $j$ with label $n$ and $m$, respectively, is estimated by

$$\hat{P}(Y_j = m | Y_i = n, (i, j) \in \mathcal{E}) \coloneqq \frac{|\{(u, v) \mid y_v = m, y_u = n, (u, v) \in \mathcal{E}\}|}{|\{(u, v) \mid y_u = n, (u, v) \in \mathcal{E}\}|} \ . \tag{2}$$

The prior distribution is also estimated from the empirical observations. We use the empirical proportion of label as a prior, i.e., $\hat{P}(Y_i = m) \coloneqq |\{u \mid y_u = m\}|/|\mathcal{V}|$. We also explore alternative designs for the likelihood and compare their performances in Section 4.2.

Note that, in implementation, all empirical distributions are computed only with the training nodes and their labels. The empirical distribution might be updated after node relabeling through the posterior computation, but we keep it the same throughout the relabeling process.

The posterior distribution can be used as a soft label to train the model, but we add uniform noise $\epsilon$ to the posterior to mitigate the risk of the posterior becoming overly confident if there are few or no neighbors. In addition, since the most probable label from the posterior might be different from the ground truth label, we interpolate the posterior with the ground truth label. To this end, we obtain the soft label $\hat{\boldsymbol{e}}_i$ of node $i$ as

$$\hat{\boldsymbol{e}}_i = (1 - \alpha)\tilde{\boldsymbol{e}}_i + \alpha \boldsymbol{e}_i \ , \tag{3}$$

where $\tilde{e}_{ik} \propto P(Y_i = k \mid \{Y_j = y_j\}_{j \in \mathcal{N}(i)}) + \beta \epsilon$. $\alpha$ and $\beta$ control the importance of interpolation and uniform noise. By enforcing $\alpha > 1/2$, we can keep the most probable label of soft label the same

as the ground truth label, but we find that this condition is not necessary in empirical experiments. We name our method as PosteL (**Poste**rior **L**abel smoothing). The detailed algorithm of PosteL is shown in Algorithm 1.

---

**Algorithm 1** PosteL: Posterior label smoothing

---

**Require:** The set of training nodes $\mathcal{V}_{\text{train}} \subset \mathcal{V}$, the number of classes $K$, one-hot encoding of training node labels $\{e_i\}_{i \in \mathcal{V}_{\text{train}}}$, and hyperparameters $\alpha$ and $\beta$.
**Ensure:** The set of soft labels $\{\hat{e}_i\}_{i \in \mathcal{V}_{\text{train}}}$
    Estimate prior distribution for $m \in [K]$: $\hat{P}(Y_i = m) = \sum_{u \in \mathcal{V}_{\text{train}}} e_{um} / |\mathcal{V}_{\text{train}}|$.
    Define the set of training neighbors for each node $u$: $\mathcal{N}_{\text{train}}(u) = \mathcal{N}(u) \cap \mathcal{V}_{\text{train}}$.
    Estimate the empirical conditional for $n, m \in [K]$:
        $\hat{P}(Y_j = m | Y_i = n, (i,j) \in \mathcal{E}) \propto \sum_{u:u \in \mathcal{V}_{\text{train}}, y_u = n} \sum_{v \in \mathcal{N}_{\text{train}}(u)} e_{vm}$.
    **for** $i \in \mathcal{V}_{\text{train}}$ **do**
        Approximate likelihood:
        $P(\{Y_j = y_j\}_{j \in \mathcal{N}_{\text{train}}(i)} | Y_i = k) \approx \prod_{j \in \mathcal{N}_{\text{train}}(i)} \hat{P}(Y_j = y_j | Y_i = k, (i,j) \in \mathcal{E})$.
        Compute posterior distribution: $P(Y_i = k \mid \{Y_j = y_j\}_{j \in \mathcal{N}_{\text{train}}(i)})$ using Equation (1).
        Add uniform noise: $\tilde{e}_{ik} \propto P(Y_i = k \mid \{Y_j = y_j\}_{j \in \mathcal{N}_{\text{train}}(i)}) + \beta \epsilon$.
        Obtain soft label: $\hat{e}_i = (1 - \alpha)\tilde{e}_i + \alpha e_i$.
    **end for**

---

## 3.2  Iterative pseudo labeling

Posterior relabeling is a method used to predict the label of a node based on the labels of its neighboring nodes. However, in transductive node classification tasks where train, validation, and test nodes coexist within the same graph, the presence of unlabeled nodes can hinder the accurate prediction of posterior labels. For instance, when a node has no labeled neighbors, the likelihood becomes one, and the posterior only relies on the prior. Moreover, in cases where labeled neighbors are scarce, noisy labels among the neighbors can significantly compromise the posterior distribution. Such challenges are particularly prevalent in sparse graphs. For example, 26.35% of nodes in the Cornell dataset have no neighbors with labels. In such scenarios, the posterior relabeling can be challenging.

To address these limitations, we propose to update the likelihoods and priors through the pseudo labels of validation and test nodes. We first train a graph neural network with the soft labels obtained via Equation (3) and predict the labels of validation and test nodes to obtain the pseudo labels. We choose the most probable label as a pseudo label from the prediction. We then update the likelihood and prior with the pseudo labels, leading to the re-calibration of the posterior smoothing and soft labels. By repeating training and re-calibration until the best validation loss of the predictor no longer decreases, we can maximize the performance of node classification. We assume that if posterior label smoothing improves classification performance with a better estimation of likelihood and prior, the pseudo labels obtained from the predictor can benefit the posterior estimation as long as there are not many false pseudo labels.

# 4  Experiments

The experimental section is composed of two parts. First, we evaluate the performance of our method for node classification through various datasets and models. Second, we provide a comprehensive analysis of our method, investigating the conditions under which it performs well and the importance of each design choice.

## 4.1  Node classification

In this section, we assess the enhancements in node classification performance across a range of datasets and backbone models. Our aim is to validate the consistent efficacy of our method across datasets and backbone models with diverse characteristics.

Table 1: Classification accuracy on 10 node classification datasets. $\Delta$ represents the performance improvement achieved by PosteL compared to the backbone model trained with the ground truth label. All results of the backbone model trained with the ground truth label are sourced from He et al. [2021].

| | Cora | CiteSeer | PubMed | Computers | Photo | Chameleon | Actor | Squirrel | Texas | Cornell |
|---|---|---|---|---|---|---|---|---|---|---|
| GCN | 87.14±1.01 | 79.86±0.67 | 86.74±0.27 | 83.32±0.33 | 88.26±0.73 | 59.61±2.21 | 33.23±1.16 | 46.78±0.87 | 77.38±3.28 | 65.90±4.43 |
| +LS | 87.77±0.97 | 81.06±0.59 | 87.73±0.24 | 89.08±0.30 | 94.05±0.26 | 64.81±1.53 | 33.81±0.75 | 49.53±1.10 | 77.87±3.11 | 67.87±3.77 |
| +KD | 87.90±0.90 | 80.97±0.56 | 87.03±0.29 | 88.56±0.36 | 93.64±0.31 | 64.49±1.38 | 33.33±0.78 | 49.38±0.64 | 78.03±2.62 | 63.61±5.57 |
| +PosteL | **88.56±0.90** | **82.10±0.50** | **88.00±0.25** | **89.30±0.23** | **94.08±0.35** | **65.80±1.23** | **35.16±0.43** | **52.76±0.64** | **80.82±2.79** | **80.33±1.80** |
| $\Delta$ | +1.42(↑) | +2.24(↑) | +1.26(↑) | +5.98(↑) | +5.82(↑) | +6.19(↑) | +1.93(↑) | +5.98(↑) | +3.44(↑) | +14.43(↑) |
| GAT | 88.03±0.79 | 80.52±0.71 | 87.04±0.24 | 83.32±0.39 | 90.94±0.68 | 63.13±1.93 | 33.93±2.47 | 44.49±0.88 | **80.82±2.13** | 78.21±2.95 |
| +LS | 88.69±0.99 | 81.27±0.86 | 86.33±0.32 | 88.95±0.31 | 94.06±0.39 | 65.16±1.49 | 34.55±1.15 | 45.94±1.60 | 78.69±4.10 | 74.10±4.10 |
| +KD | 87.47±0.94 | 80.79±0.60 | 86.54±0.31 | 88.99±0.46 | 93.76±0.31 | 65.14±1.47 | 35.13±1.36 | 43.86±0.85 | 79.02±2.46 | 73.44±2.46 |
| +PosteL | **89.21±1.08** | **82.13±0.64** | **87.08±0.19** | **89.60±0.29** | **94.31±0.31** | **66.28±1.14** | **35.92±0.72** | **49.38±1.05** | 80.33±2.62 | **80.33±1.81** |
| $\Delta$ | +1.18(↑) | +1.61(↑) | +0.04(↑) | +6.28(↑) | +3.37(↑) | +3.15(↑) | +1.99(↑) | +4.89(↑) | −0.49(↓) | +2.12(↑) |
| APPNP | 88.14±0.73 | 80.47±0.74 | 88.12±0.31 | 85.32±0.37 | 88.51±0.31 | 51.84±1.82 | 39.66±0.55 | 34.71±0.57 | 90.98±1.64 | 91.81±1.96 |
| +LS | 89.01±0.64 | 81.58±0.61 | 88.90±0.32 | 87.28±0.27 | 94.34±0.23 | **53.98±1.47** | 39.44±0.78 | **36.81±0.98** | 91.31±1.48 | 89.51±1.81 |
| +KD | 89.16±0.74 | 81.88±0.61 | 88.04±0.39 | 86.28±0.44 | 93.85±0.26 | 52.17±1.23 | **41.43±0.95** | 35.28±1.10 | 90.33±1.64 | 91.48±1.97 |
| +PosteL | **89.62±0.84** | **82.47±0.66** | **89.17±0.26** | **87.46±0.29** | **94.42±0.24** | 53.83±1.66 | 40.18±0.70 | 36.71±0.60 | **92.13±1.48** | **93.44±1.64** |
| $\Delta$ | +1.48(↑) | +2.00(↑) | +1.05(↑) | +2.14(↑) | +5.91(↑) | +1.99(↑) | +0.52(↑) | +2.00(↑) | +1.15(↑) | +1.63(↑) |
| MLP | 76.96±0.95 | 76.58±0.88 | 85.94±0.22 | 82.85±0.38 | 84.72±0.34 | 46.85±1.51 | 40.19±0.56 | 31.03±1.18 | 91.45±1.14 | 90.82±1.63 |
| +LS | 77.21±0.97 | 76.82±0.66 | 86.14±0.35 | 83.62±0.88 | 89.46±0.44 | 48.23±1.23 | 39.75±0.63 | 31.10±0.80 | 90.98±1.64 | 90.98±1.31 |
| +KD | 76.32±0.94 | 77.75±0.75 | 85.10±0.29 | 83.89±0.53 | 88.23±0.38 | 47.40±1.75 | **41.32±0.75** | 32.58±0.83 | 89.34±1.97 | 91.80±1.15 |
| +PosteL | **78.39±0.94** | **78.40±0.71** | **86.51±0.33** | **84.20±0.55** | **89.90±0.27** | **48.51±1.66** | 40.15±0.46 | **33.11±0.60** | **92.95±1.31** | **93.61±1.80** |
| $\Delta$ | +1.43(↑) | +1.82(↑) | +0.57(↑) | +1.35(↑) | +5.18(↑) | +1.66(↑) | −0.04(↓) | +2.08(↑) | +1.50(↑) | +2.79(↑) |
| ChebNet | 86.67±0.82 | 79.11±0.75 | 87.95±0.28 | 87.54±0.43 | 93.77±0.32 | 59.28±1.25 | 37.61±0.89 | 40.55±0.42 | 86.22±2.45 | 83.93±2.13 |
| +LS | 87.22±0.99 | 79.70±0.63 | 88.48±0.29 | 89.55±0.38 | 94.53±0.37 | 66.41±1.16 | 39.39±0.73 | 42.55±1.11 | 87.21±2.62 | 84.59±2.30 |
| +KD | 87.36±0.95 | 80.80±0.72 | 88.41±0.20 | 89.81±0.30 | 94.76±0.30 | 61.47±1.23 | **40.68±0.50** | 43.88±1.97 | 84.75±3.61 | 83.61±2.30 |
| +PosteL | **88.57±0.92** | **82.48±0.52** | **89.20±0.31** | **89.95±0.40** | **94.87±0.25** | **66.83±0.77** | 39.56±0.51 | **50.87±0.96** | **86.39±2.46** | **88.52±2.63** |
| $\Delta$ | +1.90(↑) | +3.37(↑) | +1.25(↑) | +2.41(↑) | +1.10(↑) | +7.55(↑) | +1.95(↑) | +10.32(↑) | +0.17(↑) | +4.59(↑) |
| GPR-GNN | 88.57±0.69 | 80.12±0.83 | 88.46±0.33 | 86.85±0.25 | 93.85±0.28 | 67.28±1.09 | 39.92±0.67 | 50.15±1.92 | 92.95±1.31 | 91.37±1.81 |
| +LS | 88.82±0.99 | 79.78±1.06 | 88.24±0.42 | 88.39±0.48 | 93.97±0.33 | 67.90±1.01 | 39.72±0.70 | 53.39±1.80 | 92.79±1.15 | 90.49±2.46 |
| +KD | **89.33±1.03** | **81.24±0.85** | 89.85±0.56 | 87.88±1.11 | 94.23±0.51 | 66.76±1.31 | **42.00±0.63** | 53.26±1.07 | **94.26±1.48** | 88.52±1.97 |
| +PosteL | 89.20±1.07 | 81.21±0.64 | **90.57±0.31** | **89.84±0.43** | **94.76±0.38** | **68.38±1.12** | 40.08±0.69 | **53.54±0.79** | 93.28±1.31 | **92.46±0.99** |
| $\Delta$ | +0.63(↑) | +1.09(↑) | +2.11(↑) | +2.99(↑) | +0.91(↑) | +1.10(↑) | +0.16(↑) | +3.39(↑) | +0.33(↑) | +1.09(↑) |
| BernNet | 88.52±0.95 | 80.09±0.79 | 88.48±0.41 | 87.64±0.44 | 93.63±0.35 | 68.29±1.58 | **41.79±1.01** | 51.35±0.73 | 93.12±0.65 | 92.13±1.64 |
| +LS | 88.80±0.92 | 80.37±1.05 | 87.40±0.38 | 88.32±0.38 | 93.70±0.21 | 69.58±0.94 | 39.60±0.53 | 52.39±0.60 | 91.80±1.80 | 90.49±1.48 |
| +KD | 87.78±0.99 | 81.20±0.86 | 87.59±0.41 | 87.35±0.40 | 93.96±0.40 | 67.75±1.42 | 41.04±0.89 | 51.25±0.83 | 93.61±1.31 | 90.33±2.30 |
| +PosteL | **89.39±0.92** | **82.46±0.67** | **89.07±0.29** | **89.56±0.35** | **94.54±0.36** | **69.65±0.83** | 40.40±0.67 | **53.11±0.87** | **93.93±1.15** | **92.95±1.80** |
| $\Delta$ | +0.87(↑) | +2.37(↑) | +0.59(↑) | +1.92(↑) | +0.91(↑) | +1.36(↑) | −1.39(↓) | +1.76(↑) | +0.81(↑) | +0.82(↑) |

**Datasets** We assess the performance of our method across 10 node classification datasets. To examine the effect of our method on diverse types of graphs, we conduct experiments on both homophilic and heterophilic graphs. Adjacent nodes in a homophilic graph are likely to have the same label. Adjacent nodes in a heterophilic graph are likely to have different labels. For the homophilic datasets, we use five datasets: the citation graphs Cora, CiteSeer, and PubMed [Sen et al., 2008, Yang et al., 2016], and the Amazon co-purchase graphs Computers and Photo [McAuley et al., 2015]. For the heterophilic datasets, we use five datasets: the Wikipedia graphs Chameleon and Squirrel [Rozemberczki et al., 2021], the Actor co-occurrence graph Actor [Tang et al., 2009], and the webpage graphs Texas and Cornell [Pei et al., 2020]. Detailed statistics of each dataset are illustrated in Appendix A.

**Experimental setup and baselines** We evaluate the performance of PosteL across various backbone models, ranging from MLP, which ignores underlying structure between nodes, to six widely used graph neural networks: GCN [Kipf and Welling, 2016], GAT [Veličković et al., 2017], APPNP [Gasteiger et al., 2018], ChebNet [Defferrard et al., 2016], GPR-GNN [Chien et al., 2020], and BernNet [He et al., 2021]. We follow the experimental setup and backbone implementations of He et al. [2021]. Specifically, we use fixed 10 train, validation, and test splits with ratios of 60%/20%/20%, respectively, and measure the accuracy at the lowest validation loss. We report the mean performance and 95% confidence interval. The model is trained for 1,000 epochs, and we apply early stopping when validation loss does not decrease during the last 200 epochs. For all models, the learning rate is validated within $\{0.001, 0.002, 0.01, 0.05\}$, and weight decay within $\{0, 0.0005\}$. The search spaces of the other model-dependent hyperparameters are provided in Appendix B. We validate two hyperparameters for PosteL: posterior label ratio $\alpha \in \{0.1, 0.2, 0.3, 0.4, 0.5, 0.6, 0.7, 0.8, 0.9, 1.0\}$ and uniform noise ratio $\beta \in \{0, 0.1, 0.2, 0.3, 0.4, 0.5, 0.6, 0.7, 0.8, 0.9\}$.

We compare our method with two different soft labeling methods, including label smoothing (LS) [Szegedy et al., 2016b] and knowledge distillation (KD) [Hinton et al., 2015]. For KD, we

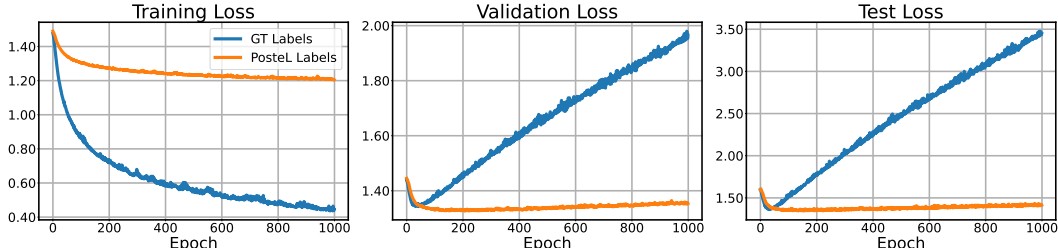

Figure 2: Loss curve of GCN trained on PosteL labels and ground truth labels on the Squirrel dataset.

use an ensemble of average logits from three independently trained GNNs as a teacher model. The temperature parameter for KD is set to four following the previous work [Stanton et al., 2021].

**Results**  In Table 1, the classification accuracy and 95% confidence interval for each of the seven models across the 10 datasets are presented. In most cases, PosteL outperforms baseline methods across various settings, demonstrating significant performance enhancements and validating its effectiveness for node classification. Specifically, our method performs better in 67 cases out of 70 settings against the ground truth labels. Furthermore, among these settings, 39 cases show improvements over the 95% confidence interval. Notably, on the Cornell dataset with the GCN backbone, our method achieves a substantial performance enhancement of 14.43%. When compared to the other soft label methods, PosteL performs better in most cases as well. The knowledge distillation method shows comparable performance with the GPR-GNN baseline, but even in this case, there are marginal differences between the two approaches.

## 4.2 Analysis

In this section, we analyze the main experimental result from various perspectives, including design choices, ablations, and computational complexity.

**Learning curves analysis**  We investigate the influence of soft labels on the learning dynamics of GNNs by visualizing the loss function of GCNs with and without soft labels. Figure 2 visualizes the differences between training, validation, and test losses with and without the PosteL labels on the Squirrel dataset. From the training loss, we observe that the cross entropy with the PosteL labels converges to a higher loss than that with the ground truth labels. The curve shows that predicting soft labels is more difficult than predicting ground truth labels. On the other hand, the validation and test losses with the soft labels converge to lower losses than those with the ground truth labels. Especially, up to 200 epochs, we observe that no overfitting happens with the soft labels. We conjecture that predicting the correct PosteL label implies the correct prediction of the local neighborhood structure since the PosteL labels contain the local neighborhood information of the target node. Hence, the model trained with PosteL labels could have a better understanding of the graph structure, potentially leading to a better generalization performance. A similar context prediction approach has been proposed as a pertaining method in previous studies [Hu et al., 2019, Rong et al., 2020]. We provide the same curves for all datasets in Figure 6 and Figure 7 in Appendix D. All curves across all datasets show similar patterns.

**Influence of neighborhood label distribution**  Our approach assumes that the distribution of neighborhood labels varies depending on the label of the target node. If there are no significant differences between the neighborhood's label distributions, the posterior relabeling assigns similar soft labels for all nodes, making our method similar to the uniform noise method.

Figure 3 shows the neighborhood label distribution for three different datasets. In the PubMed and Texas datasets, we observe a notable difference in the conditionals when w.r.t the different labels of a target node. The PubMed dataset is known to be homophilic, where nodes with the same labels are likely to be connected, and the conditional distributions match the characteristics of the homophilic dataset. The Texas dataset, a heterophilic dataset, shows that some pairs of labels more frequently appear in the graph. For example, when the target node has the label of 1, their neighborhoods will likely have the label of 5. On the other hand, the conditionals of the Actor dataset do not vary much

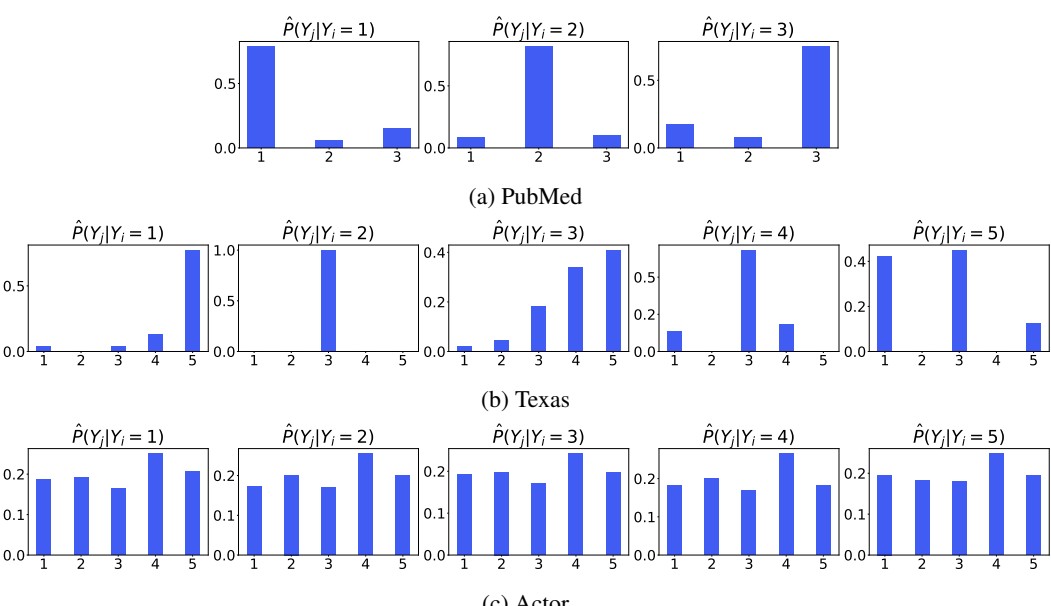

(a) PubMed

(b) Texas

(c) Actor

Figure 3: Empirical conditional distributions between two adjacent nodes. We omit the adjacent condition $(i, j) \in \mathcal{E}$ from the figures for simplicity.

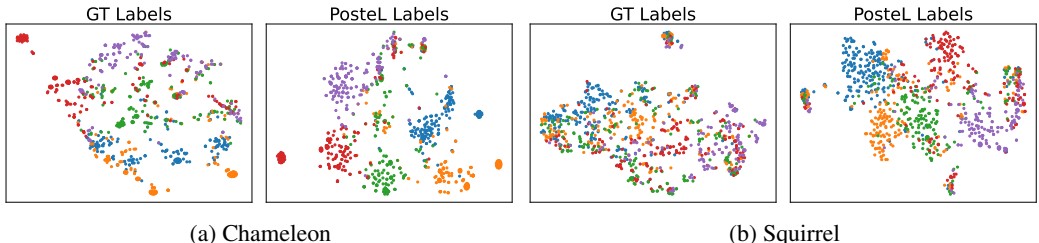

(a) Chameleon

(b) Squirrel

Figure 4: t-SNE plots of the final layer representation of the Chameleon and Squirrel datasets. For each dataset, the left figure displays the representations trained on the ground truth labels, while the right figure displays the representations trained on the PosteL labels.

regarding the label of the target node. In such a case, the prior will likely dominate the posterior. Therefore, the posterior may not provide useful information about neighborhood nodes, potentially limiting the effectiveness of our method. This analysis aligns with the results in Table 1, where the improvement of the Actor dataset is less significant than those of the PubMed and Texas datasets. The neighborhood label distributions for all datasets are provided in Figure 8 and Figure 9 in Appendix E.

**Visualization of node embeddings** Figure 4 presents the t-SNE [Van der Maaten and Hinton, 2008] plots of node embeddings from the GCN with the Chameleon and Squirrel datasets. The node color represents the label. For each dataset, the left plot visualizes the embeddings with the ground truth labels, while the right plot visualizes the embeddings with PosteL labels. The visualization shows that the embeddings from the soft labels form tighter clusters compared to those trained with the ground truth labels. This visualization results coincide with the t-SNE visualization of the previous work of Müller et al. [2019].

**Effect of iterative pseudo labeling** We evaluate the impact of iterative pseudo labeling by analyzing the loss curve at each iteration. Figure 5 illustrates the loss curves for different iterations on the Cornell dataset. As the iteration progresses, the validation and test losses after 1,000 epochs keep decreasing. In this example, the model performs best after four iteration steps. We find that the best validation performance is obtained from 1.13 iterations on average. We provide the average iteration steps in Appendix C used to report the results in Table 1.

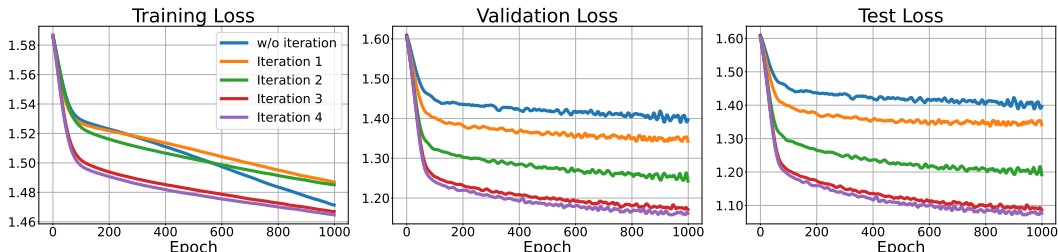

Figure 5: The impact of the iterative pseudo labeling: loss curves of GCN on the Cornell dataset.

Table 2: Classification accuracy with various choices of likelihood model. PosteL (local-1) and (local-2) indicate that the likelihood is estimated within one- and two-hop neighbors of a target node, respectively. PosteL (norm.), shortened from PosteL (normalized), indicates that the likelihood is normalized based on the degree of a node.

|  | Cora | CiteSeer | Computers | Photo | Chameleon | Actor | Texas | Cornell |
|---|---|---|---|---|---|---|---|---|
| GCN | 87.14±1.01 | 79.86±0.67 | 83.32±0.33 | 88.26±0.73 | 59.61±2.21 | 33.23±1.16 | 77.38±3.28 | 65.90±4.43 |
| +PosteL (local-1) | 88.26±1.07 | 81.42±0.46 | 89.08±0.31 | 93.61±0.40 | 65.36±1.25 | 33.48±1.03 | 79.02±3.11 | 71.97±4.10 |
| +PosteL (local-2) | 88.62±0.97 | 81.92±0.42 | 88.62±0.48 | 93.95±0.37 | 65.10±1.55 | 34.63±0.46 | 78.20±2.79 | 73.28±4.10 |
| +PosteL (norm.) | **89.00±0.99** | 81.86±0.70 | **89.30±0.39** | **94.13±0.39** | **66.00±1.14** | 34.90±0.63 | 80.33±2.95 | 80.00±1.97 |
| +PosteL | 88.56±0.90 | **82.10±0.50** | **89.30±0.23** | 94.08±0.35 | 65.80±1.23 | **35.16±0.43** | **80.82±2.79** | **80.33±1.80** |

**Design choices of likelihood model** We explore various valid design choices for likelihood models. We introduce two variants of PosteL: PosteL (normalized) and PosteL (local-$H$). In Equation (2), each edge has an equal contribution to the conditional. The conditional can be influenced by a few numbers of nodes with many connections. To mitigate the importance of high-degree nodes, we alternatively test the following conditional, denoted as PosteL (normalized):

$$\hat{P}^{\text{norm.}}(Y_j = m | Y_i = n, (i,j) \in \mathcal{E}) := \frac{\sum_{y_u = n} \sum_{v \in \mathcal{N}(u)} \frac{1}{|\mathcal{N}(u)|} \cdot \mathbb{1}[y_v = m]}{|\{y_u = n \mid u \in \mathcal{V}\}|} \, ,$$

where $\mathbb{1}$ is an indicator function.

In PosteL (local-$H$), we estimate the likelihood and prior distributions of each node from their respective $H$-hop ego graphs. Specifically, the likelihood of PosteL (local-$H$) is formulated as follows:

$$\hat{P}^{\text{local-}H}(Y_j = m | Y_i = n, (i,j) \in \mathcal{E}) := \frac{|\{(u,v) | y_v = m, y_u = n, (u,v) \in \mathcal{E}, u, v \in \mathcal{N}^{(H)}(i)\}|}{|\{(u,v) | y_u = n, (u,v) \in \mathcal{E}, u, v \in \mathcal{N}^{(H)}(i)\}|} \, ,$$

where $\mathcal{N}^{(H)}(i)$ denotes the set of neighborhoods of node $i$ within $H$ hops. Through the local likelihood, we test the importance of global and local statistics in the smoothing process.

Table 2 shows the comparison between these variants. The likelihood with global statistics, e.g., PosteL and PosteL (normalized), performs better than the local likelihood methods, e.g., PosteL (local-1) and PosteL (local-2) in general, highlighting the importance of simultaneously utilizing global statistics. Especially in the Cornell dataset, a significant performance gap between PosteL and PosteL (local) is observed. PosteL (normalized) demonstrates similar performance to PosteL.

**Ablation studies** To highlight the importance of each component in PosteL, we perform ablation studies on three components: posterior smoothing without uniform noise (PS), uniform smoothing (UN), and iterative pseudo labeling (IPL). Table 3 presents the performance results from the ablation studies.

The configuration with all components included achieves the highest performance, underscoring the significance of each component. The iterative pseudo labeling proves effective across almost all datasets, with a particularly notable impact on the Cornell dataset. However, even without iterative pseudo labeling, the performance remains competitive, suggesting that its use can be decided based on available resources. Additionally, incorporating uniform noise into the posterior distribution enhances performance on several datasets. Moreover, PosteL consistently outperforms the approach using only uniform noise, a widely used label smoothing method.

Table 3: Ablation studies on three main components of PosteL on GCN. PS stands for posterior label smoothing without uniform noise, UN stands for uniform noise added to the posterior distribution, and IPL stands for iterative pseudo labeling. We use ✓ to indicate the presence of the corresponding component in training and ✗ to indicate its absence. IPL with one indicates the performance with a single pseudo labeling step.

| PS | UN | IPL | Cora | CiteSeer | Computers | Photo | Chameleon | Actor | Texas | Cornell |
|---|---|---|---|---|---|---|---|---|---|---|
| ✗ | ✗ | ✗ | $87.14_{\pm1.01}$ | $79.86_{\pm0.67}$ | $83.32_{\pm0.33}$ | $88.26_{\pm0.73}$ | $59.61_{\pm2.21}$ | $33.23_{\pm1.16}$ | $77.38_{\pm3.28}$ | $65.90_{\pm4.43}$ |
| ✓ | ✗ | ✗ | $88.11_{\pm1.22}$ | $80.95_{\pm0.52}$ | $88.86_{\pm0.40}$ | $93.55_{\pm0.30}$ | $64.53_{\pm1.23}$ | $33.48_{\pm0.62}$ | $78.52_{\pm2.46}$ | $68.52_{\pm4.43}$ |
| ✗ | ✓ | ✗ | $87.77_{\pm0.97}$ | $81.06_{\pm0.59}$ | $89.08_{\pm0.30}$ | $94.05_{\pm0.26}$ | $64.81_{\pm1.53}$ | $33.81_{\pm0.75}$ | $77.87_{\pm3.11}$ | $67.87_{\pm3.77}$ |
| ✓ | ✗ | ✓ | $\mathbf{88.56_{\pm0.90}}$ | $81.64_{\pm0.57}$ | $88.70_{\pm0.27}$ | $93.70_{\pm0.37}$ | $64.25_{\pm1.93}$ | $34.71_{\pm0.76}$ | $\mathbf{80.82_{\pm2.79}}$ | $80.16_{\pm1.97}$ |
| ✓ | ✓ | ✗ | $87.83_{\pm0.92}$ | $82.09_{\pm0.44}$ | $89.17_{\pm0.31}$ | $93.98_{\pm0.34}$ | $\mathbf{66.19_{\pm1.60}}$ | $34.91_{\pm0.48}$ | $79.51_{\pm3.61}$ | $71.97_{\pm5.25}$ |
| ✓ | ✓ | 1 | $87.96_{\pm0.90}$ | $\mathbf{82.33_{\pm0.52}}$ | $89.16_{\pm0.30}$ | $94.06_{\pm0.27}$ | $65.89_{\pm1.51}$ | $34.96_{\pm0.48}$ | $80.16_{\pm2.79}$ | $\mathbf{80.33_{\pm1.97}}$ |
| ✓ | ✓ | ✓ | $\mathbf{88.56_{\pm0.90}}$ | $82.10_{\pm0.50}$ | $\mathbf{89.30_{\pm0.23}}$ | $\mathbf{94.08_{\pm0.35}}$ | $65.80_{\pm1.23}$ | $\mathbf{35.16_{\pm0.43}}$ | $\mathbf{80.82_{\pm2.79}}$ | $\mathbf{80.33_{\pm1.80}}$ |

Table 4: Accuracy of the model trained with sparse labels. The ratio indicates the percentage of nodes used for training.

| | ratio | Cora | CiteSeer | Computers | Photo | Chameleon | Actor | Texas | Cornell |
|---|---|---|---|---|---|---|---|---|---|
| GCN | 5% | $80.03_{\pm0.57}$ | $70.19_{\pm0.49}$ | $85.32_{\pm0.60}$ | $92.39_{\pm0.24}$ | $45.96_{\pm2.48}$ | $25.20_{\pm0.83}$ | $54.23_{\pm6.35}$ | $\mathbf{50.58_{\pm5.84}}$ |
| +PosteL | | $\mathbf{80.42_{\pm0.64}}$ | $\mathbf{71.08_{\pm0.65}}$ | $\mathbf{86.22_{\pm0.45}}$ | $\mathbf{92.66_{\pm0.21}}$ | $\mathbf{51.35_{\pm1.19}}$ | $\mathbf{27.04_{\pm0.51}}$ | $\mathbf{57.52_{\pm1.97}}$ | $50.36_{\pm3.43}$ |
| GCN | 10% | $83.05_{\pm0.51}$ | $72.09_{\pm0.46}$ | $86.68_{\pm0.59}$ | $92.49_{\pm0.29}$ | $51.55_{\pm1.67}$ | $26.78_{\pm0.68}$ | $60.08_{\pm2.56}$ | $53.64_{\pm3.49}$ |
| +PosteL | | $\mathbf{83.50_{\pm0.36}}$ | $\mathbf{73.76_{\pm0.26}}$ | $\mathbf{87.47_{\pm0.37}}$ | $\mathbf{92.88_{\pm0.30}}$ | $\mathbf{56.33_{\pm1.86}}$ | $\mathbf{28.07_{\pm0.19}}$ | $\mathbf{61.63_{\pm2.87}}$ | $\mathbf{57.75_{\pm1.86}}$ |
| GCN | 20% | $84.46_{\pm0.68}$ | $73.93_{\pm0.69}$ | $87.12_{\pm0.33}$ | $93.24_{\pm0.33}$ | $55.57_{\pm1.18}$ | $27.42_{\pm0.76}$ | $63.33_{\pm2.05}$ | $52.91_{\pm2.65}$ |
| +PosteL | | $\mathbf{85.32_{\pm0.65}}$ | $\mathbf{75.73_{\pm0.39}}$ | $\mathbf{87.77_{\pm0.19}}$ | $\mathbf{93.47_{\pm0.18}}$ | $\mathbf{60.91_{\pm1.07}}$ | $\mathbf{29.23_{\pm0.50}}$ | $\mathbf{64.87_{\pm2.74}}$ | $\mathbf{56.92_{\pm2.39}}$ |
| GCN | 30% | $85.76_{\pm0.46}$ | $75.56_{\pm0.44}$ | $87.02_{\pm0.49}$ | $93.14_{\pm0.27}$ | $59.41_{\pm1.08}$ | $28.81_{\pm0.50}$ | $65.64_{\pm4.36}$ | $60.40_{\pm3.96}$ |
| +PosteL | | $\mathbf{86.04_{\pm0.37}}$ | $\mathbf{77.30_{\pm0.65}}$ | $\mathbf{88.09_{\pm0.31}}$ | $\mathbf{93.47_{\pm0.27}}$ | $\mathbf{63.64_{\pm0.98}}$ | $\mathbf{30.21_{\pm0.39}}$ | $\mathbf{69.80_{\pm3.86}}$ | $\mathbf{64.95_{\pm2.08}}$ |
| GCN | 40% | $\mathbf{86.32_{\pm0.43}}$ | $77.17_{\pm0.52}$ | $87.88_{\pm0.58}$ | $93.76_{\pm0.20}$ | $60.44_{\pm1.20}$ | $29.71_{\pm0.72}$ | $67.88_{\pm2.47}$ | $62.00_{\pm2.12}$ |
| +PosteL | | $86.23_{\pm0.37}$ | $\mathbf{79.22_{\pm0.32}}$ | $\mathbf{88.21_{\pm0.29}}$ | $\mathbf{93.99_{\pm0.24}}$ | $\mathbf{63.82_{\pm1.44}}$ | $\mathbf{31.05_{\pm0.40}}$ | $\mathbf{73.76_{\pm2.59}}$ | $\mathbf{67.41_{\pm4.71}}$ |

**Complexity analysis**  The computational complexity of calculating the posterior label is $O(|\mathcal{E}|K)$. Since the labeling is performed before the learning stage, the time required to process the posterior label can be considered negligible. The training time increases linearly w.r.t the number of iterations with the pseudo labeling. However, experiments show that an average of 1.13 iterations is needed, making our approach feasible without having too many iterations. The proof of computational complexity is in Appendix C.

### 4.3  Training with sparse labels

Our method relies on global statistics estimated from training nodes. However, in scenarios where training data is sparse, the estimation of global statistics can be challenging. To assess the effectiveness of the label smoothing from graphs with sparse labels, we conduct experiments with varying sizes of a training set. We vary the size of the training set from 5% to 40% of an entire dataset and conduct the classification experiments with the same setting used in the previous section. The percentage of validation nodes is set to 20% for all experiments. Table 4 provides the classification performance with sparse labels. Even in scenarios with sparse labels, PosteL consistently outperforms models trained on ground truth labels in most cases. These results show that our method can effectively capture global statistics even when training data is limited.

## 5  Conclusion

In this paper, we proposed a novel posterior label smoothing method, PosteL, designed to enhance node classification performance in graph-structured data. Our approach integrates both local neighborhood information and global label statistics to generate soft labels, thereby improving generalization and mitigating overfitting. Extensive experiments across various datasets and models demonstrated the effectiveness of PosteL, showing significant performance gains compared to baseline methods despite its simplicity.

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

## A   Dataset statistics

We provide detailed statistics about the dataset used for the experiments in Table 5.

| Dataset | # nodes | # edges | # features | # classes |
|---|---|---|---|---|
| Cora | 2,708 | 5,278 | 1,433 | 7 |
| CiteSeer | 3,327 | 4,552 | 3,703 | 6 |
| PubMed | 19,717 | 44,324 | 500 | 3 |
| Computers | 13,752 | 245,861 | 767 | 10 |
| Photo | 7,650 | 119,081 | 745 | 8 |
| Chameleon | 2,277 | 31,396 | 2,325 | 5 |
| Actor | 7,600 | 30,019 | 932 | 5 |
| Squirrel | 5,201 | 198,423 | 2,089 | 5 |
| Texas | 183 | 287 | 1,703 | 5 |
| Cornell | 183 | 277 | 1,703 | 5 |

Table 5: Statistics of the dataset utilized in the experiments.

## B   Detailed experimental setup

In this section, we provide the computer resources and search space for model hyperparameters. Our experiments are executed on AMD EPYC 7513 32-core Processor and a single NVIDIA RTX A6000 GPU with 48GB of memory. We use the same model hyperparameter search space as He et al. [2021]. Specifically, we set the number of layers for all models to two. The dropout ratio for the linear layers is fixed at 0.5. For the GCN [Kipf and Welling, 2016], the hidden layer dimension is set to 64. The GAT [Veličković et al., 2017] uses eight heads, each with a hidden dimension of eight. For the APPNP [Gasteiger et al., 2018], a two-layer MLP with a hidden dimension of 64 is used, the power iteration step is set to 10, and the teleport probability is chosen from $\{0.1, 0.2, 0.5, 0.9\}$. For the MLP, the hidden dimension is set to 64. For the ChebNet [Defferrard et al., 2016], the hidden dimension is set to 32, and two propagation steps are used. For the GPR-GNN [Chien et al., 2020], a two-layer MLP with a hidden dimension of 64 is used as the feature extractor neural network, and the random walk path length is set to 10. The PPR teleport probability is chosen from $\{0.1, 0.2, 0.5, 0.9\}$. For BernNet [He et al., 2021], a two-layer MLP with a hidden dimension of 64 is used as the feature extractor, and the polynomial approximation order is set to 10. The dropout ratio for the propagation layers in both GPR-GNN and BernNet is chosen from $\{0.0, 0.1, 0.2, 0.3, 0.4, 0.5, 0.6, 0.7, 0.8, 0.9\}$.

## C   Complexity analysis

In this section, we provide a detailed analysis of the time complexity of Section 3.1. Specifically, we demonstrate the time complexity of obtaining the prior and likelihood distributions separately. Finally, we determine the time complexity of computing the posterior distribution using these distributions.

First, the prior distribution $\hat{P}(Y_i = m)$ can be obtained as follows:

$$\hat{P}(Y_i = m) = \frac{|\{u \mid y_u = k\}|}{|\mathcal{V}|} = \frac{\sum_{u \in \mathcal{V}} e_{um}}{|\mathcal{V}|}. \tag{4}$$

The time complexity of calculating Equation (4) is $O(|\mathcal{V}|)$, so the time complexity of calculating the prior distribution for $K$ classes is $O(|\mathcal{V}|K)$.

Next, calculating the empirical conditional $\hat{P}(Y_j = m|Y_i = n, (i, j) \in \mathcal{E})$ from Equation (2) can be performed as follows:

$$\hat{P}(Y_j = m|Y_i = n, (i, j) \in \mathcal{E}) \propto \sum_{u:u \in \mathcal{V}, y_u = n} \sum_{v \in \mathcal{N}(u)} e_{vm}. \tag{5}$$

Table 6: Average iteration counts of iterative pseudo labeling for each backbone and dataset used to report Table 1.

| | Cora | CiteSeer | PubMed | Computers | Photo | Chameleon | Actor | Squirrel | Texas | Cornell |
|---|---|---|---|---|---|---|---|---|---|---|
| GCN+PosteL | 2.5 | 2.2 | 1.5 | 1 | 0.9 | 0.9 | 1.1 | 0.7 | 1.8 | 2.5 |
| GAT+PosteL | 1.6 | 1.8 | 1 | 1.2 | 0.7 | 0.8 | 2 | 1.1 | 3.1 | 2.4 |
| APPNP+PosteL | 1.9 | 2 | 1.1 | 0.8 | 1.1 | 1 | 1.1 | 0.9 | 1.4 | 2.9 |
| MLP+PosteL | 1.7 | 2.2 | 0.4 | 0.7 | 0.7 | 0.1 | 0.8 | 0.6 | 0.9 | 2.4 |
| ChebNet+PosteL | 1.6 | 2.1 | 1.2 | 0.6 | 0.6 | 1 | 0.7 | 0.7 | 2 | 2 |
| GPR-GNN+PosteL | 0.8 | 1.1 | 0.8 | 0.5 | 1.3 | 1 | 0.3 | 0.7 | 1.1 | 1 |
| BernNet+PosteL | 1.5 | 1.8 | 0.9 | 0.8 | 1 | 1.5 | 1.5 | 0.5 | 1.2 | 2.1 |

The time complexity of calculating Equation (5) for all possible pairs of $m$ and $n$ is $O(\sum_{u \in \mathcal{V}} |\mathcal{N}(u)|K)$. Since $\sum_{u \in \mathcal{V}} \mathcal{N}(u) = 2|\mathcal{E}|$, the time complexity for calculating empirical conditional is $O(|\mathcal{E}|K)$.

The likelihood is approximated through the product of empirical conditional distributions, denoted as $P(\{Y_j = y_j\}_{j \in \mathcal{N}(i)} | Y_i = k) \approx \prod_{j \in \mathcal{N}(i)} \hat{P}(Y_j = y_j | Y_i = k, (i,j) \in \mathcal{E})$. Likelihood calculation for all training nodes operates in $O(\sum_{u \in \mathcal{V}} |\mathcal{N}(u)|K)$ time complexity. So the overall computational complexity for likelihood calculation is $O(|\mathcal{E}|K)$.

After obtaining the prior distribution and likelihood, the posterior distribution is obtained by Bayes' rule in Equation (1). Applying Bayes' rule for $|\mathcal{V}|$ nodes and $K$ classes can be done in $O(|\mathcal{V}|K)$. So the overall time complexity is $O((|\mathcal{E}| + |\mathcal{V}|) K)$. In most cases, $|\mathcal{V}| < |\mathcal{E}|$, so the time complexity of PosteL is $O(|\mathcal{E}|K)$.

In Section 3.2, iterative pseudo labeling is proposed, which involves iteratively refining the pseudo labels of validation and test nodes to calculate posterior labels. Since this process requires training the model from scratch for each iteration, the number of iterations can be a significant bottleneck in terms of runtime. Consequently, the iteration counts are evaluated to assess this aspect. The mean iteration counts for each backbone and dataset in Table 1 are summarized in Table 6. With an overall mean iteration count of 1.13, we argue that this level of additional time investment is justifiable for the sake of performance enhancement.

## D  Learning curves analysis for all datasets

The learning curves for all datasets are provided in Figure 6 and Figure 7.

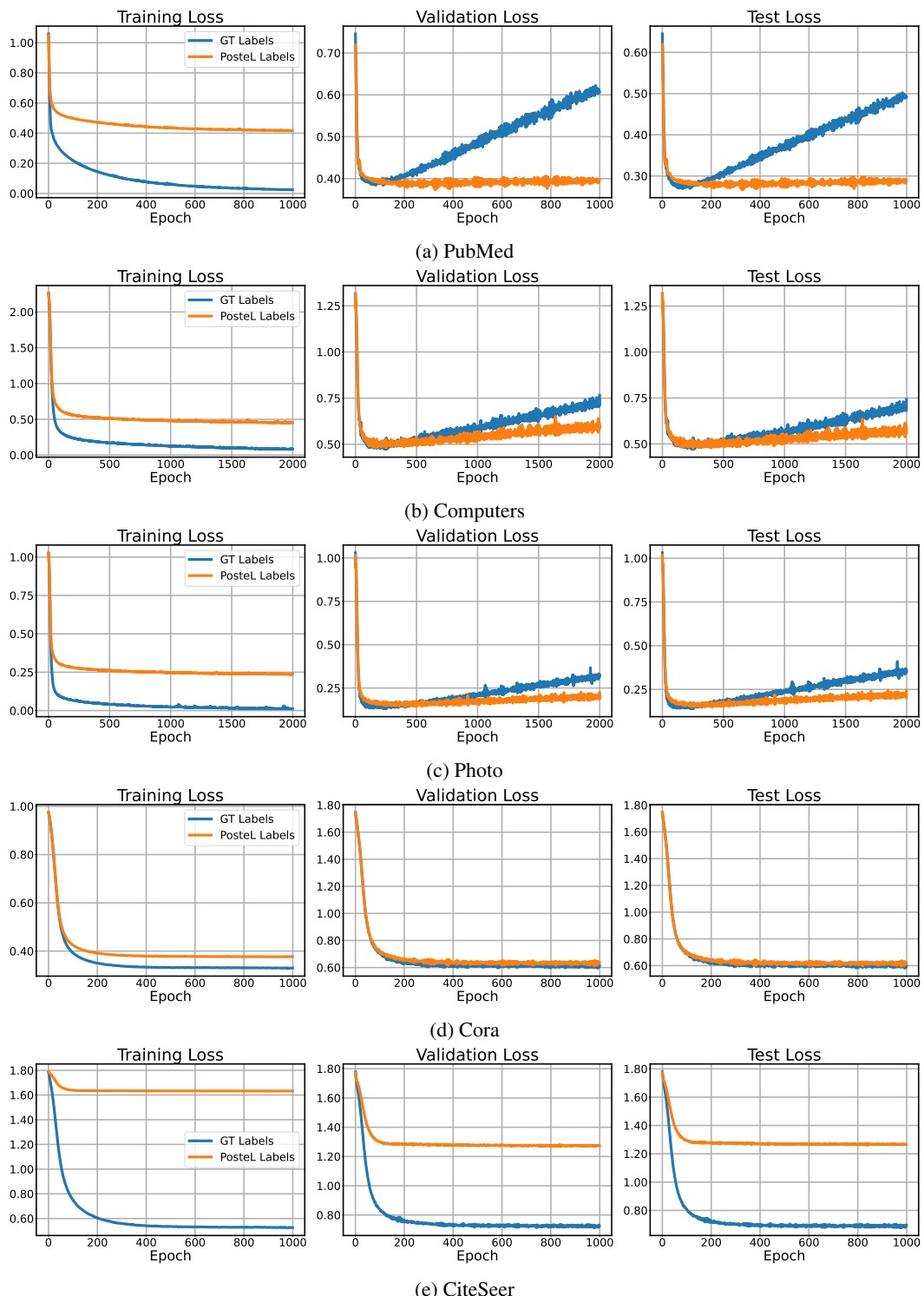

Figure 6: Loss curve of GCN trained on PosteL labels and ground truth labels on homophilic datasets.

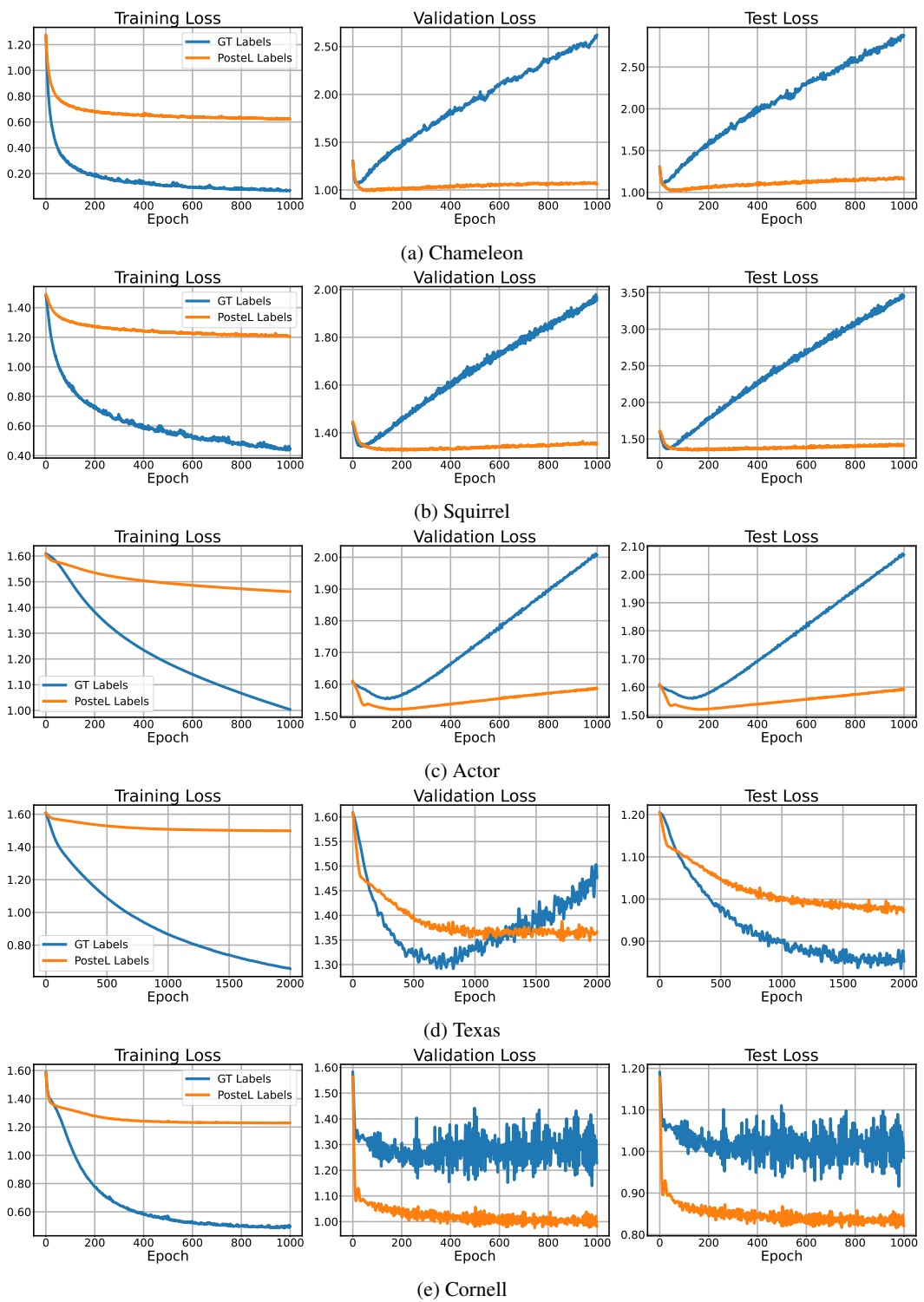

Figure 7: Loss curve of GCN trained on PosteL labels and ground truth labels on heterophilic datasets.

# E    Empirical conditional distribution for all datasets

The empirical conditional distribution for all datasets is provided in Figure 8 and Figure 9.

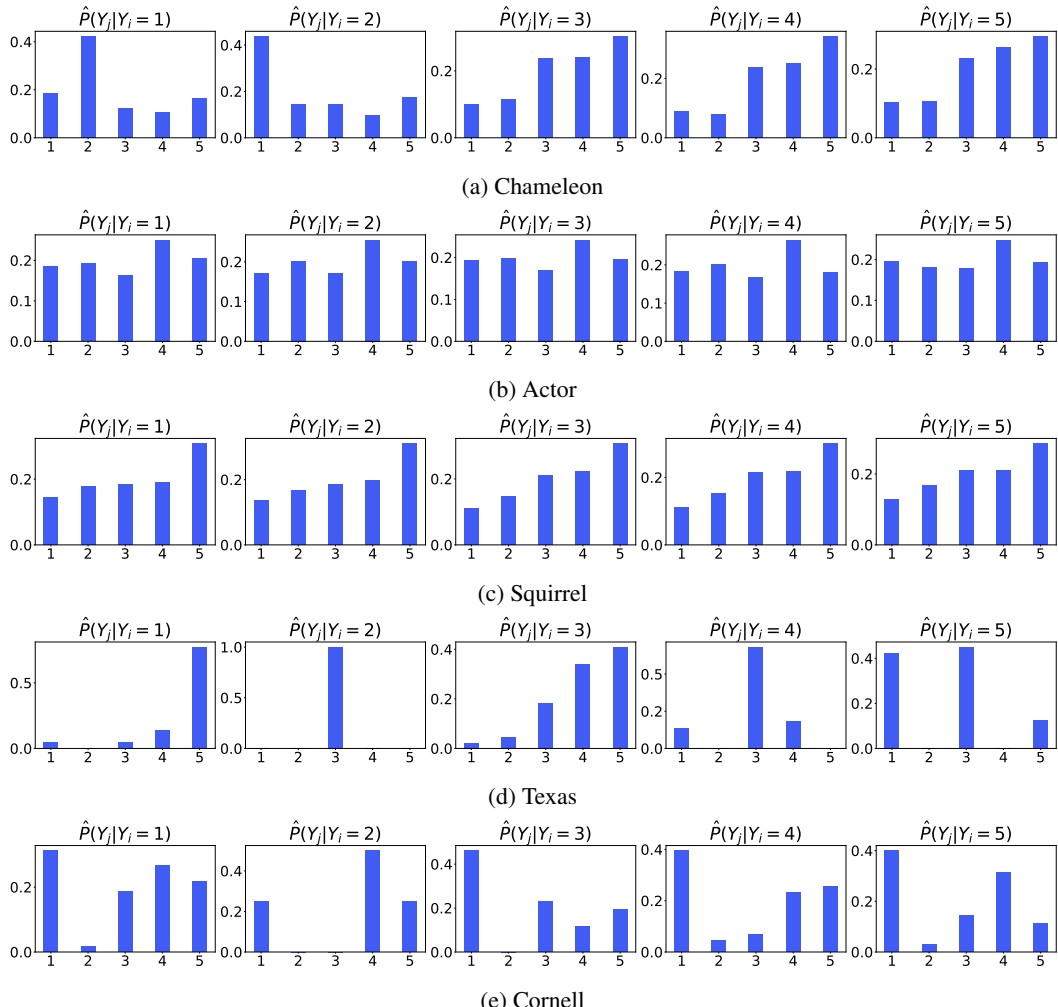

Figure 8: Empirical conditional distributions between two adjacent nodes on heterophilic graphs.

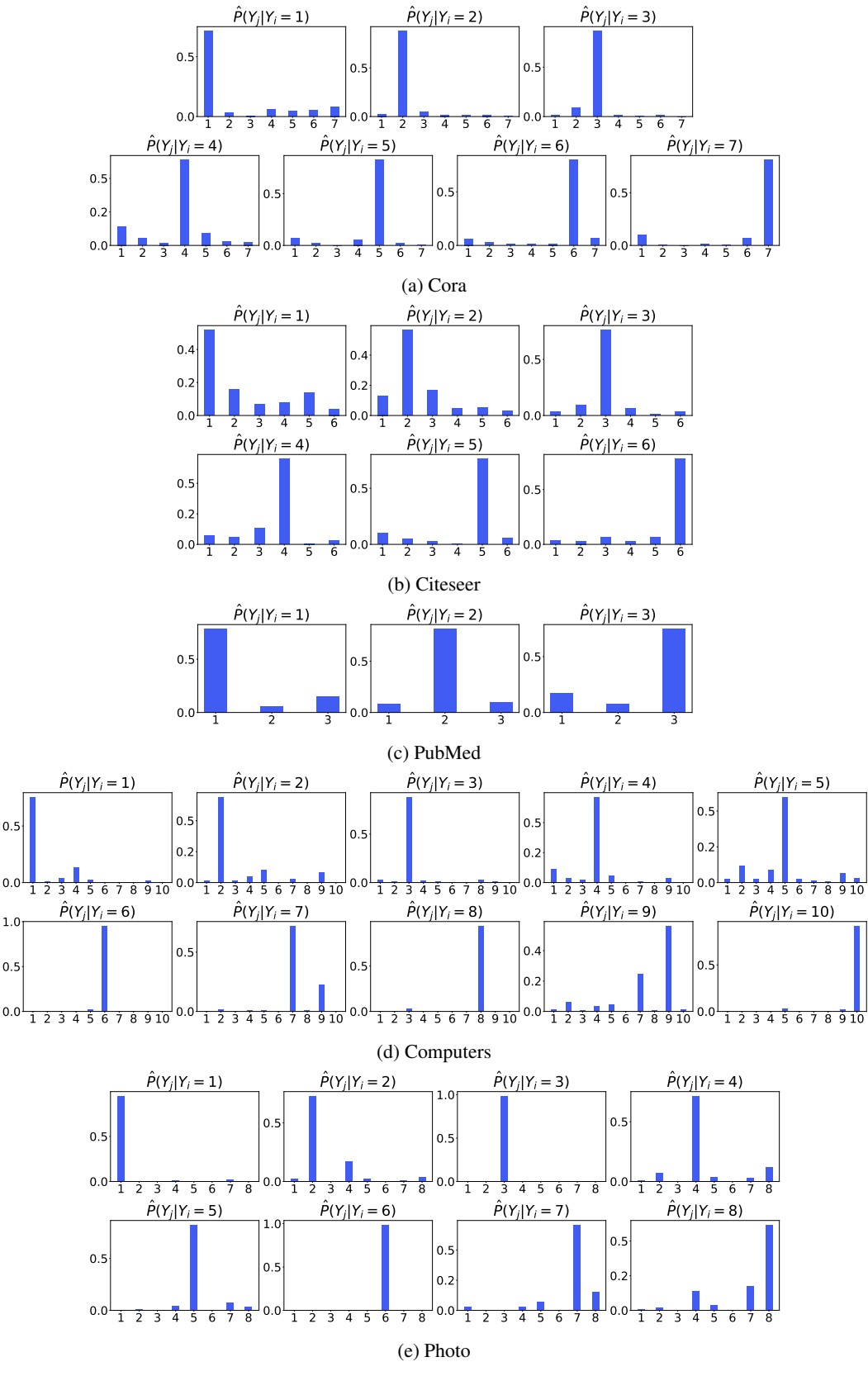

Figure 9: Empirical conditional distributions between two adjacent nodes on homophilic graphs.

