# OpenReview forum: "Posterior Label Smoothing for Node Classification"
_NeurIPS.cc/2024/Conference — Submitted to NeurIPS 2024_

### Official Review · Reviewer_k4jx · 2024-06-29

**Soundness:** 2
**Presentation:** 2
**Contribution:** 1
**Rating:** 2
**Confidence:** 4

**Summary:**

This paper proposed label-smoothing to improve the transductive node classification in GNN.

**Strengths:**

Label-smoothing and knowledge distillation are applied for node classification performance.

**Weaknesses:**

1.	The paper could benefit from discussing related works that combine label-smoothing with Graph Neural Networks (GNNs), such as [1] and [2]. Including these would provide a more comprehensive context for the current research.

2.	The proposed method lacks a theoretical motivation or analysis. Providing this would strengthen the paper's scientific rigor and help readers better understand the underlying principles.

3.	The proposed method bears similarities to the approach in [1]. A direct comparison with this work would clarify the novel contributions of the current study and situate it within the existing research landscape.

4.	Iterative pseudo-labeling is a well-established technique in the field. The paper should address this, explaining how the current application differs from or builds upon previous uses of this method.
Addressing these points could significantly enhance the paper's depth and impact.

---
[1]: Wang, Y., Cai, Y., Liang, Y., Wang, W., Ding, H., Chen, M., ... & Hooi, B. (2021). Structure-aware label smoothing for graph neural networks. arXiv preprint arXiv:2112.00499.

[2]: Zhang, Wentao, et al. "Node dependent local smoothing for scalable graph learning." Advances in Neural Information Processing Systems 34 (2021): 20321-20332.

**Questions:**

See weaknesses

**Limitations:**

The limitation should be clarified in the paper. The current limitations are not clear.

---

> ### Author Rebuttal · Authors · 2024-08-06
>
> We sincerely appreciate your effort in the review process and your constructive feedback to improve our paper. We particularly recognize the issue of lacking related work. We have conducted additional experiments and strengthened the related work section. Detailed responses to your questions are provided below.
>
> **W1 & W3: Related work that combine label-smoothing with GNNs**
>
> We have addressed this question in the general response. Please refer to general response 1. In short, we confirm that our method outperforms two recently proposed methods [1,2].
>
>
> **W2: Lack of theoretical motivation or analysis**
>
> We agree that we lack theoretical analysis for our method, and providing it would enhance the soundness and contribution of our work. In this response, we try to make a connection between our method and a previous theoretical analysis on label smoothing. In [3], the authors identify that uniform label smoothing can help generalization when the training label is noisy. In this sense, our method can also be seen as a mitigation of the noisy label. In traditional classification tasks where the relation between data points are unclear, it would be difficult to develop a meaningful smoothing. Whereas, in node classification tasks, we know that connected nodes are related to each other and therefore develop more informative smoothing method as we have done. More rigorous analysis in this theoretical direction would be a promising future work.
>
> On the other hand, as Reviewer `yTqG` mentioned, our main contribution lies in the strong empirical results and diverse analyses we present. We believe that these contributions also offer positive contribution to the community.
>
> **W4: Explaining how the current application differs from or builds upon previous uses of iterative-pseudo labeling**
>
> The main difference between existing iterative pseudo labeling methods and ours lies in the use of the pseudo labels. As [4] noted, traditional methods “essentially adopt the semi-supervised strategy”, while we adopt the supervised settings. In more detail, once the pseudo labels are generated, traditional methods incorporate them into training set to retrain the model [4, 5], whereas we use the pseudo label to update the neigborhood and conditional statistics used for smoothing.
>
> [1] Wang, Yiwei, et al. "Structure-aware label smoothing for graph neural networks." arXiv preprint (2021).
> [2] Zhou, Kaixiong, et al. "Adaptive label smoothing to regularize large-scale graph training." SDM 2023.
> [3] Chen, Blair, et al. "An investigation of how label smoothing affects generalization." arXiv preprint  (2020).
> [4] Cascante-Bonilla, Paola, et al. "Curriculum labeling: Revisiting pseudo-labeling for semi-supervised learning." Proceedings of the AAAI conference on artificial intelligence. Vol. 35. No. 8. 2021.
> [5] Xu, Qiantong, et al. "Iterative pseudo-labeling for speech recognition." Interspeech 2020.

---

> > ### Comment · Reviewer_k4jx · 2024-08-07
> > **Feedback**
> >
> > I appreciate the authors' response to our comments. However, upon careful consideration, I find that the proposed method shares significant similarities with the work presented in [1]. As a result, the primary contribution of this study appears to be limited to additional empirical observations. Given these factors, I maintain our original assessment and score.
> >
> > [1] Wang, Yiwei, et al. "Structure-aware label smoothing for graph neural networks." arXiv preprint (2021).

---

> > > ### Author Response · Authors · 2024-08-08
> > >
> > > We thank the reviewer for a prompt response. While we do not request a change to the reviewer’s assessment, we do ask for the following that can assist other reviewers's fair assessment, as the current response lacks details.
> > >
> > > As we have answered in the general response, we believe that our work is clearly different from [1] since [1] is designed under homophilic assumption, whereas our work is based on the assumption that the neighborhood label distribution should differ when the node label differs. The difference is reflected in the results with heterophilic datasets, and we do believe this is a significant difference. If [1] is similar to our work, could the reviewer elaborate on which aspects our approach is similar to [1]?
> > >
> > > Just for reviewers' information, we also want to highlight that [1] is still in preprint and has never been published yet as far as we aware. We found some ambiguities in Equation 4 in their paper, where $c$ is used to define $\eta$, there is no information provided on the left side of the equation while implementing their work. We implemented the work based on our educated guesses, but the reproducibility of the work is limited.
> > >
> > > [1] Wang, Yiwei, et al. "Structure-aware label smoothing for graph neural networks." arXiv preprint (2021).

---

### Official Review · Reviewer_Qv2Y · 2024-07-06

**Soundness:** 3
**Presentation:** 3
**Contribution:** 2
**Rating:** 7
**Confidence:** 3

**Summary:**

The paper proposes PosteL, a label smoothing method utilizing posterior distribution for node classification in graph-structured data. It is basically a preprocessing method for GNNs, generating soft labels based on neighborhood context and global label statistics before the training phase.

**Strengths:**

1. The paper is generally well-written and easy to follow. For example, Fig. 1 is clear and intuitive.
2. The method is simple yet effective. PosteL can be combined seamlessly with existing methods.
3. The results are significant. PosteL is tested on seven neural network models across ten datasets, demonstrating significant improvements in classification accuracy.

**Weaknesses:**

1. The authors select $\alpha$ and $\beta$ from a wide range but did not explore the parameter sensitivity of PosteL. The sensitivity to hyperparameters could be a potential limitation, necessitating careful tuning, which may reduce the credibility of the experiments.
2. The authors do not seem to clarify the difference between PosteL and other label smoothing methods for node classification (or methods that can be adapted to node classification), which makes the novelty of the method unclear. The paper could explore other smoothing techniques or baselines in more depth for a comprehensive comparison.

**Questions:**

1. Are there any examples to support the assumption that the neighborhood labels are conditionally independent given the label of the node to be relabeled (line 28)? It would be helpful for reader understanding to include this in the paper.
2. Why are the node features not considered in the relabeling process?
3. The authors demonstrate the effectiveness of PosteL with sparse labels, but the reasons behind are lacking explanation. Also, how does the performance compare to other preprocessing methods?

**Limitations:**

The authors discuss the case when the prior likely dominates the posterior, which limits the effectiveness of the proposed PosteL.

---

> ### Author Rebuttal · Authors · 2024-08-06
>
> We sincerely appreciate your effort in the review process and recognition of the strengths of our work. We have carefully considered all of your comments, and detailed responses to your questions are provided below. We hope this helps address any concerns you may have.
>
> **W1: Sensitivity analysis of the hyperparameters $\alpha$ and $\beta$**
>
> We have addressed this question in the general response. Please refer to general response 2. In short, we confirm that our method is insensitive to $\alpha$ and $\beta$.
>
> **W2: Comprehensive comparison with related work**
>
> We have addressed this question in the general response. Please refer to general response 1. In short, we confirm that our method outperforms two recently proposed methods.
>
> **Q1: Are there examples to support the assumption that the neighborhood labels are conditionally independent given the label of the node?**
>
> Unfortunately, we couldn't find any empirical evidence on our claim about the independency. Having said that, this kind of assumption is widely adopted in machine learning in general to make the model simple and tractable. One example is the i.i.d. assumption between data points, which is widely accepted even though it doesn't make sense in reality. In the context of graph learning, [1,2,3] assume that L-hop ego graph of a target node is i.i.d., which is similar to our assumption.
>
> **Q2: Why are the node features not considered in the relabeling process?**
>
> KD can be seen as a relabeling process that considers the node features, since teacher models generate soft labels using the node features as input. Our experimental results show that PosteL, which focuses on utilizing label information rather than node features, outperforms KD. Incorporating both node features and labels into the relabeling process would be an interesting idea, but it may hurt our core contribution 'simple but effective'. Having said that this could be an interesting research direction for future work.
>
> **Q3: How does the performance on sparse label compare to other preprocessing methods?**
>
> We conduct additional experiments to evaluate the performance of label smoothing baselines on sparse labels. PosteL shows the best performance, achieving the highest results in 22 out of 24 cases. Label smoothing with uniform noise (LS) is less affected by sparse labels since it does not rely on labeled neighborhoods, performing slightly better than when using GT labels. In contrast, SALS [4] and ALS [5] are significantly affected by sparse labels because they depend on labeled neighborhoods.
>
> |         | Ratio | Cora  | CiteSeer | Computers | Photo | Chameleon | Actor | Texas | Cornell |
> |---------|-------|-------|----------|-----------|-------|-----------|-------|-------|---------|
> | GCN     | 5%    | 80.0±0.6 | 70.2±0.5 | 85.3±0.6 | 92.4±0.2 | 46.0±2.5 | 25.2±0.8 | 54.2±6.4 | 50.6±5.8 |
> | +LS     |       | 80.2±0.7 | 70.0±0.6 | 85.9±0.7 | 92.6±0.3 | 47.8±2.0 | 24.6±0.5 | 52.8±7.2 | **53.2±3.9** |
> | +SALS   |       | 80.2±0.9 | 70.0±0.7 | 86.0±0.5 | 92.4±0.2 | 43.7±2.4 | 25.3±0.6 | 53.4±6.4 | 52.3±5.0 |
> | +ALS    |       | **80.4±0.9** | 70.0±0.6 | 85.3±0.5 | 92.5±0.2 | 47.0±1.6 | 24.9±0.6 | 53.1±6.1 | 50.8±3.4 |
> | +PosteL |       | **80.4±0.6** | **71.1±0.7** | **86.2±0.5** | **92.7±0.2** | **51.4±1.2** | **27.0±0.5** | **57.5±2.0** | 50.4±3.4 |
> | GCN     | 10%   | 83.1±0.5 | 72.1±0.5 | 86.7±0.6 | 92.5±0.3 | 51.6±1.7 | 26.8±0.7 | 60.1±2.6 | 53.6±3.5 |
> | +LS     |       | 83.1±0.3 | 72.0±0.3 | 87.3±0.4 | **92.9±0.3** | 52.7±0.8 | 26.6±0.6 | 59.9±3.0 | 55.8±2.7 |
> | +SALS   |       | 83.1±0.6 | 72.3±0.5 | 87.3±0.3 | 92.5±0.4 | 50.2±2.1 | 26.2±0.9 | 60.6±2.3 | 55.6±2.8 |
> | +ALS    |       | 83.1±0.4 | 72.2±0.4 | 86.4±0.6 | 92.8±0.3 | 52.3±1.7 | 26.5±0.8 | **61.6±2.3** | 54.1±3.3 |
> | +PosteL |       | **83.5±0.4** | **73.8±0.3** | **87.5±0.4** | **92.9±0.3** | **56.3±1.9** | **28.1±0.2** | **61.6±2.9** | **57.8±1.9** |
> | GCN     | 20%   | 84.5±0.7 | 73.9±0.7 | 87.1±0.3 | 93.2±0.3 | 55.6±1.2 | 27.4±0.8 | 63.3±2.1 | 52.9±2.7 |
> | +LS     |       | 84.6±0.6 | 74.3±0.5 | 87.6±0.2 | 93.4±0.2 | 58.9±0.9 | 27.7±0.7 | 62.4±2.1 | 54.7±2.2 |
> | +SALS   |       | 84.9±0.8 | 74.0±0.6 | 87.6±0.3 | 93.1±0.3 | 55.6±2.0 | 28.0±1.0 | 62.9±2.7 | 52.1±2.3 |
> | +ALS    |       | 84.7±0.7 | 74.1±0.6 | 87.2±0.4 | 93.2±0.2 | 57.5±1.5 | 28.1±0.4 | 62.7±2.1 | 55.6±2.1 |
> | +PosteL |       | **85.3±0.7** | **75.7±0.4** | **87.8±0.2** | **93.5±0.2** | **60.9±1.1** | **29.2±0.5** | **64.9±2.7** | **56.9±2.4** |
>
>
> **Q3: The reasons behind the effectiveness of PosteL with sparse labels. are lacking explanation.**
>
> The reason for our concern regarding PosteL in sparse label settings is that PosteL's advantage comes from utilizing information from labeled neighborhoods. However, in sparse label settings, there are fewer labeled neighborhoods. We believe the pseudo labeling process mitigates this concern, allowing PosteL to work effectively in sparse label settings.
>
> SALS and ALS face the same issue as PosteL, but they do not propose a method to mitigate it. Consequently, we observe that SALS and ALS often perform worse than when using GT labels.
>
> [1] Verma, Saurabh, and Zhi-Li Zhang. "Stability and generalization of graph convolutional neural networks." KDD 2019.
> [2] Garg, Vikas, Stefanie Jegelka, and Tommi Jaakkola. "Generalization and representational limits of graph neural networks." ICML 2020.
> [3] Wu, Qitian, et al. "Handling distribution shifts on graphs: An invariance perspective." arXiv preprint 2022.
> [4] Wang, Yiwei, et al. "Structure-aware label smoothing for graph neural networks." arXiv preprint arXiv:2112.00499 (2021).
> [5] Zhou, Kaixiong, et al. "Adaptive label smoothing to regularize large-scale graph training." SDM 2023.

---

> > ### Comment · Reviewer_Qv2Y · 2024-08-12
> >
> > Thanks for your response, I keep the original score unchanged.

---

### Official Review · Reviewer_ESPd · 2024-07-14

**Soundness:** 2
**Presentation:** 3
**Contribution:** 2
**Rating:** 5
**Confidence:** 4

**Summary:**

This paper introduces Posterior Label smoothing (PosteL), an innovative approach to enhance node classification on graph-structured data. PosteL integrates local neighborhood information with global label statistics to generate soft labels, aiming to improve model generalization and mitigate overfitting. The authors demonstrate the effectiveness of PosteL through extensive experiments on various datasets and models, showing significant performance improvements over baseline methods.

**Strengths:**

1.	The paper is well-written and easy to follow.
2.	The authors provide a comprehensive set of experiments across different datasets and models, which substantiates the effectiveness of the proposed method.
3.	The figures and tables are well-organized, clear and easy to understand.
4.	The method is relatively lightweight and easy to implement at the technical level.

**Weaknesses:**

1.	While the paper mentions the computational complexity, a deeper analysis or comparison with existing methods could provide more insight. For example, maybe you could provide some compared experiments with existing methods on time/resource consumption.
2.	The reliance on global label statistics might introduce bias in cases where the dataset has inherent class imbalance or label noise.
3.	The article "Rethinking the inception architecture for computer vision" appears twice in your reference list; please consolidate these entries. Carefully review your references to maintain standardization.
4.	The author compares two soft label methods that were proposed quite some time ago (from 2015 and 2016, respectively). Are there any experimental results comparing with more recent methods? Otherwise, the persuasiveness of the experiments might not be so strong.
5.	Please maintain consistent terminology throughout the text. The term "over-fitting" in line 46 should be changed to "overfitting" to be consistent with the rest of the context.
6.	Authors could provide more details on the sensitivity analysis of the hyperparameters α and β, which are crucial for the method's performance.

**Questions:**

1.	Could the authors provide more details on the implementation of the iterative pseudo labeling process?
2.	In the iterative pseudo labeling process, the validation and test sets are introduced to training. Whether the model will be overfitting to validation and test sets?
3.	What strategies were used to select the hyperparameters α and β, and how sensitive is the model performance to these choices?
4.	In Figures 6 and 7, it can be observed that not all the validation and test losses with the soft labels converge to lower losses than those with the ground truth labels. Does this indicate that the method has overly mitigated overfitting to the data?
5.	Regarding the overfitting effect mentioned in "Learning curves analysis," I am curious whether other soft label based methods can achieve the same effect, or is this a unique effect of the method presented in this paper?

**Limitations:**

Yes

---

> ### Author Rebuttal · Authors · 2024-08-06
>
> We sincerely appreciate your effort in the review process and your constructive feedback. We have carefully reviewed our paper based on your comments, and detailed responses to your questions are provided below. Finally, thank you for highlighting the editorial issues regarding the duplicated reference and inconsistent notation. We will address these issues in a future revision.
>
> **W1: Comparison with time/resource consumption**
>
> We estimated training time of PosteL and the other baselines and presented in the following table. With IPL, PosteL requires more training time, being 1.3 times slower than ALS [1] and 5.7 times slower than using GT labels. If this computational overhead is too heavy, we can use PosteL without IPL or IPL with one iteration as an alternative. PosteL without IPL is 2 times faster than KD and ALS, and PosteL with IPL with one iteration is also faster than KD and ALS while not sacrificing the accuracy. We reported the accuracy of each variation in Table 3 of our paper. Additionally, we measured the memory consumption of PosteL on the Cora dataset. PosteL requires only 0.87MB of additional memory to process labels.
>
> ||Vanilla|LS|KD|SALS|ALS|PosteL|PosteL w/o IPL|PosteL with one iteration IPL|
> |-|-|-|-|-|-|-|-|-|
> |Time(s)|0.9|0.7|3.5|0.7|3.9|5.1|1.6|3.1|
>
> **W2: The reliance on global label statistics might introduce bias in cases where the dataset has inherent class imbalance or label noise.**
>
> - [Class imbalance] We believe that the class imbalance would not be a problem if the training and test sets follow the same underlying distribution. In fact, some datasets we used have class imbalance property, such as Computers and Texas, where our method improves the accuracy. In the table below, we provide the proportion of nodes for each class across different datasets.
>
> ||1|2|3|4|5|6|7|8|9|10|
> |-|-|-|-|-|-|-|-|-|-|-|
> |Computers (%)|3.1|15.5|10.2|3.9|37.5|2.2|3.5|5.9|15.6|2.1|
> |Texas (%)|18.0|0.5|9.8|55.1|16.3|
>
> - [Noisy label] To check the robustness of our method under noisy label, we conducted additional experiments, where 30% of the training data labels were randomly flipped to different labels. The table below shows the results on noisy labels. We observed that PosteL still outperforms GT on noisy labels.
>
> ||Cora|CiteSeer|Pubmed|Computers|Photo|Chameleon|Actor|Squrrel|Texas|Cornell|
> |-|-|-|-|-|-|-|-|-|-|-|
> |GCN|84.7±1.2|77.5±0.7|85.7±0.4|87.9±0.3|92.9±0.4|58.2±0.8|31.5±0.8|41.5±0.9|65.4±7.3|56.7±5.2|
> |+PosteL|87.9±1.0|81.0±0.5|87.2±0.2|88.0±0.2|93.2±0.4|59.4±1.4|33.4±1.8|43.1±1.0|79.8±2.3|72.9±15.2|
>
> **W4: Experimental results comparing with recent methods**
>
> We have addressed this question in the general response. Please refer to general response 1. In short, we confirm that our method outperforms two recently proposed methods [1,2].
>
> **W6 & Q3: Sensitivity analysis of the hyperparameters $\alpha$ and $\beta$**
>
> We have addressed this question in the general response. Please refer to general response 2. In short, we confirm that our method is insensitive to $\alpha$ and $\beta$.
>
> **Q1: Details on the implementation of the IPL process**
>
> Details on the implementation of the iterative pseudo labeling are provided below.
> 1. Generate soft labels for the **training nodes** by applying Equation 1 to the **GT labels of the training nodes**.
> 2. Train the GNN using the **soft labels of training nodes** obtained from Step 1.
> 3. Using the trained GNN from Step 2, obtain pseudo labels for the **validation and test nodes**.
> 4. Using the **GT labels of the training nodes and the pseudo-labels of the validation and test nodes**, generate new soft labels for the **training nodes** by applying Equation 1.
> 5. Train a new GNN using the soft labels of the **training nodes** obtained from Step 4.
> 6. Repeat Steps 4 and 5 until the validation loss does not decrease.
>
> **Q2: In the IPL process, the validation and test sets are introduced to training. Whether the model will be overfitting to validation and test sets?**
>
> We apologize for confusion. As described in the answer to the previous question, the model is not trained to fit the pseudo labels of validation and test nodes directly. Instead, pseudo labels are used to calculate statistical values that update the soft labels of training nodes.
>
> **Q4: Not all the validation and test losses with the soft labels converge to lower losses than those with the ground truth labels.**
>
> Indeed, this is an interesting observation. We found that even when the validaiton loss with the soft labels is greater than that of the GT labels, the accuracy of the PosteL is better than the model trained on GT labels. Due to the characteristics of the cross entropy loss, two different prediction results with the same label after argmax can have very different level of losses, e.g., for binary prediction, $\arg\max([0,1])=\arg\max([0.4,0.6])=1$.
> This can be understood as an overfitting mitigation, but we believe it is not 'overly' mitigated since the PosteL is still effective in terms of the accuracy. Similar observation is noted in previous work as well [3].
>
> **Q5: Loss curve of other soft label based methods**
>
> In Figure 2 of attached [PDF](https://openreview.net/attachment?id=wvWDdogE8H&name=pdf) file in the general response, we plot learning curves of different models with their best hyperparameter configurations. PosteL shows the smallest gap between training and testing losses compared to other methods, showing strong empirical evidence on overfitting alleviation. Furthermore, the validation and test losses of our methods still remains at the similar level while those of other methods are keep increasing when the training is longer.
>
> [1] Zhou, Kaixiong, et al. "Adaptive label smoothing to regularize large-scale graph training." SDM 2023.
> [2] Wang, Yiwei, et al. "Structure-aware label smoothing for graph neural networks." arXiv preprint (2021).
> [3] Müller, Rafael, Simon Kornblith, and Geoffrey E. Hinton. "When does label smoothing help?." NeurIPS 2019.

---

> > ### Comment · Reviewer_ESPd · 2024-08-11
> >
> > Authors partially addressed some of my concerns. I have raised my score.

---

### Official Review · Reviewer_yTqG · 2024-07-29

**Soundness:** 3
**Presentation:** 3
**Contribution:** 3
**Rating:** 6
**Confidence:** 4

**Summary:**

This work proposes a preprocessing step to refine labels of nodes in a structured graph that can benefit different graph-related transductive classification tasks. Inspired by the success of label smoothing in other machine learning tasks, the authors propose a label smoothing procedure based on a Bayesian inference that aggregates local and global information to estimate the soft labels. The procedure consists of mixing the soft and hard labels and an iterative regime akin to the Bayes update, which makes the method adaptive to different regularities present in different datasets. Authors conduct experiments applied to various models and datasets to support the efficacy of their methodology. They also provide an ablation study and further analyses of the results that shed light on different aspects of their proposal.

**Strengths:**

1. The identified gap is relevant, and applying label smoothing to the context of graph node classification bears novelty in terms of its application in this context.
2. The empirical results suggest that the proposed solution addresses the research question successfully and merits the attention of the community.
3. Moreover, the core ideas are communicated clearly and coupled with intuitive illustrations demonstrating the proposed method, which is very well appreciated.
4. And lastly, the results and analyses are communicated well.

**Weaknesses:**

1. **Related work**: Currently, the related works seem to provide references to earlier studies that, for the most part, motivate this work and are not methodologically close to it. For example, there is no reference to closely related works that either adopted label smoothing or conducted a very similar procedure in the context of graph data. Most notable is "[Adaptive Label Smoothing To Regularize Large-Scale Graph Training](https://epubs.siam.org/doi/abs/10.1137/1.9781611977653.ch7)" which appears to have a very similar procedure but a different approach to obtain the soft labels.

Moreover, the current statements imply that the current work is the first to suggest label smoothing for the graph data. To be more concrete, line 81 needs to be expanded, and more closely related works need to be discussed. For example, to compare the current approach and highlight similarities and key distinctions with earlier works that are closely related to it.

Some other related works could be the following:
- [Structure-Aware Label Smoothing for Graph Neural Networks](https://arxiv.org/abs/2112.00499)
- [Label Efficient Regularization and Propagation for Graph Node Classification](https://ieeexplore.ieee.org/abstract/document/10234505)
- [Node Dependent Local Smoothing for Scalable Graph Learning](https://proceedings.neurips.cc/paper_files/paper/2021/hash/a9eb812238f753132652ae09963a05e9-Abstract.html)


2. **Design decisions and theory**: besides complexity analysis, the study could have been accompanied by convergence analysis and more theoretically founded justification. However, this does not reduce the value of the work as its empirical results provide a strong signal for the effectiveness of the method, which merits future work toward theoretical assessment and explanation of its success.

3. **Background information**: the classification task that uses the preprocessed smooth labels is not defined explicitly, which makes the work less accessible for the readers without prior knowledge.

4. **The IPL step** is proposed to address the presence of "unlabeled nodes"; however, it is hard to follow how the varying training size experiment reported in Table 4 is analogous to the unlabeled node scenario. Perhaps it is due to a lack of background information mentioned in point 3.

5. **Suggestions to rephrase**:
   line 236: "mitigate the importance", perhaps some rephrasing is needed.
   line 189: "learning curve" -> "loss curve"
   line 209: "when", some rephrasing might be needed

**Questions:**

1. Can you please elaborate on the similarities and distinctions of your work with [Adaptive Label Smoothing To Regularize Large-Scale Graph Training](https://epubs.siam.org/doi/abs/10.1137/1.9781611977653.ch7)?

2. line 159, the experimental setup: it is a bit concerning that according to Figures 2, 6, and 7, the baselines appear to have a chance if the right regularization (other than label smoothing) and/or optimal early stopping is applied. What is your thought on that? Also, the weight decay grid search is very limited; is there any reason for that? Moreover, do you have a justification as to why we should not take the early stopping as a hyperparameter as opposed to the current setup that is fixed for each variation (baseline, +LS, +KD, +PosteL)?

3. Table 1: some explanation as to why certain models, such as GPR-GNN or BernNet, appear to not be benefiting from label smoothing.

4. line 117: it is mentioned that the condition is unnecessary based on empirical results. Could you please refer to the experiment in which this is being observed?

5. line 109: is this still true in the case of the IPL? In the IPL, the empirical distributions seem to be updated according to the pseudo-labels on the validation/test, correct?

6. Figures 2, 6, and 7: are they based on one split? Perhaps it would be useful to look at the mean and variance across all splits, similar to the reporting in Table 1.

**Limitations:**

Limitations are addressed in the body of the text. It is perhaps preferable to have the important limitations mentioned in a separate section or in the conclusion as well.

---

> ### Author Rebuttal · Authors · 2024-08-06
>
> Thank you for your dedicated effort in the review process. We appreciate the constructive feedback and your recognition of the strengths of our work. We have carefully considered all the points you mentioned and provided detailed responses to each question below. Additionally, we will address the editorial issues you mentioned in the revised version.
>
> **W1 & Q1: Related work on label smoothing for node classification**
>
> We have addressed this question in the general response. Please refer to general response 1. In short, we confirm that our method outperforms two recently proposed methods.
>
> **W2: Theoretical contribution**
>
> We agree that we lack theoretical analysis for our method, and providing it would enhance the soundness and contribution of our work. [1] shows the effect of label smoothing with uniform noise on the view of generalization, which can be a potential direction for future works.
>
> **W3 & W4: Experimental details and the meaning of experiments with varying proportion of training set.**
>
> We first sorry for the use of a jargon. In this work, we tackle a node classification problem in a *transductive* setting. In transductive node classification, a graph and the labels of some nodes are given as a training set, and we predict the labels of the remaining nodes as a test set. Hence, in training time, we can observe the entire connectivity between all nodes including a test set. A small portion of training set can be used as a validation set.
>
> Table 4 in our paper shows the changes in prediction performance when the proportion of labeled nodes increases. Intuitively, our method may not work well when the number of nodes with known label is small, but through the experiments, we found that our method works surprisingly well with the limited amount of training labels. Although it needs to be investigated more thoroughly, we conjecture that the estimation of the global label statistics could help to compensate the influence of sparse neighbors.
>
> **Q2: It is a bit concerning that according to Figures 2, 6, and 7, the baselines appear to have a chance if the right regularization and/or optimal early stopping is applied. Also, the weight decay grid search is very limited; is there any reason for that? Moreover, do you have a justification as to why we should not take the early stopping as a hyperparameter as opposed to the current setup that is fixed for each variation?**
>
> In Figure 2 of the attached [PDF] file in the general response, we plot the learning curves of different models with their best hyperparameter configurations within our grid search space.
> PosteL shows the smallest gap between training and testing losses compared to other methods, showing strong empirical evidence on overfitting alleviation. Furthermore, the validation and test losses of our methods still remains at the similar level while those of other methods are keep increasing when the training is longer.
>
> The grid search spaces are originally proposed in the BernNet paper [2]. We follow the same procedure for a fair comparison. The early stopping strategy is also adopted from the BernNet. Our empirical observation with Figure 2 in the general response also suggest that 200 epochs are sufficient to check potential improvement in training.
>
> **Q3: Table 1: some explanation as to why certain models, such as GPR-GNN or BernNet, appear to not be benefiting from label smoothing**
>
> We respectfully disagree that our method does not benefit from label smoothing for certain models. For GPR-GNN and BernNet, our method improves performance in 19 out of 20 cases, with 8 cases showing significant improvement over a 95% confidence interval. The relatively smaller improvements for heterophilic graphs compared to other backbone GNNs might be because GPR-GNN and BernNet are already known to process heterophilic graphs well, thus leaving less room for additional gains.
>
> **Q4: line 117: it is mentioned that the condition is unnecessary based on empirical results. Could you please refer to the experiment in which this is being observed?**
>
> Sorry for the confusion. The empirical results are observed while we conducted experiments but never shown in the submitted manuscript. For the information, we provide the hyperparameter sensitivity analysis in the general response by varying the values of $\alpha$ and $\beta$.
>
> In Figure 1 of the attached [PDF] file in the general response, the blue line represents the performance with varying $\alpha$, while the green line shows the performance with varying $\beta$. The red dotted line indicates the performance with the GT labels. We found that the model performance is quite robust to the choice of hyperparameter except for extreme cases.
>
> **Q5: line 109: is this still true in the case of the IPL? In the IPL, the empirical distributions seem to be updated according to the pseudo-labels on the validation/test, correct?**
>
> Correct. We keep the original empirical distributions the same to make the IPL simple. We conducted experiments with the updated empirical distributions but could not find a significant gain in results.
>
> **Q6: Figures 2, 6, and 7: are they based on one split? Perhaps it would be useful to look at the mean and variance across all splits, similar to the reporting in Table 1.**
>
> Yes, Figures 2, 6, and 7 are based on a single split. Figure 2 in the general response shows the loss curve averaged over 10 splits on the Squirrel dataset. We will update these figures in future revision.
>
> [1] Chen, Blair, et al. "An investigation of how label smoothing affects generalization." arXiv preprint  (2020).
> [2] He, Mingguo, Zhewei Wei, and Hongteng Xu. "Bernnet: Learning arbitrary graph spectral filters via bernstein approximation." NeurIPS 2021.
>
> [PDF]: https://openreview.net/attachment?id=wvWDdogE8H&name=pdf

---

> ### Comment · Reviewer_yTqG · 2024-08-12
> **Follow up comments and questions on the response of the authors**
>
> I thank the authors for their diligence in addressing the concerns and questions.
> Q4, Q5, and Q6 are addressed, and I have no further inquiry.
>
> **Regarding Q1**, the explanation provided is the type of information expected to be included in the paper. For the final revision of the paper, it is highly recommended that similarities be included and that the existence of [1] be acknowledged. Likewise, it is important to highlight the key differences between each step in as much detail as possible. ALS's label propagation is similar to posterior label smoothing in this work, with the key difference being that the prior in the former is the hard labels while in the latter is the global statistics, which appears to be one of the elements that accommodate both homophilic and heterophilic graphs. A similar comparison is expected for the "label refinement" and "smooth pacing" steps from ALS. It would be great if the authors also provided such an explanation in their answer to this inquiry, which could further assist reviewers in their final opinion.
>
> Also, in Table 1 of the attached PDF, it would be great to highlight which datasets are homophilic and heterophilic.
>
>
> **Concerning the response to W3 & W4**, while the current experiment conducted under section 4.3 is a valid experiment, it addresses the question of the efficiency of the model in learning. The title of the section, however, sets the expectation for the reader to see the performance of the model against different label sparsity ratios.
>
> Here are some possibilities:
> - a) varying node label sparsity across the graph but with a fixed training set size.
> - b) varying node label sparsity across train and test sets. (so the sparsity is different per each set)
> - c) fixed label sparsity across the graph, but varying training set size while test set size is fixed.
>
> The authors provide results for "case c", while "case a" seems to assess the robustness of the method against different sparsities because it addresses sparsity more directly, while "case c" includes the confounding effect of both train size and label sparsity.
>
> My recommendation is to rephrase the current section title, and if authors intend to report robustness against sparsity, "case a" is a better option. If they decide not to include such an experiment in their work, it is then suggested to make necessary clarifications in the title and body of the section corresponding to this experiment, and they may mention this as a future work.
>
>
> **Concerning Q2**,
> First, a follow-up question:
> Loss curves for GT in Figure 2 from the paper and Figure 2 from the attached PDF do not match, e.g., GT labels in 1000 epochs get minimized up to ~0.4 loss, while in the new attached PDF for the same number of epochs, it gets minimized up to ~0.8. Similarly, for the PosteL, e.g., the difference between validation and test is more pronounced based on Figure 2 of the paper than the figure in the attachment.
>
> I was expecting them to be the same. Why are they different?
>
> Second, I rephrase my question as I feel the main point is still not fully addressed:
> According to the original Figure 2 from the paper, early stopping at epoch ~50 provides a similar performance to Postel labels.
>
> That holds true for all datasets in Figure 6 and for some datasets in Figure 7. So, the question is, given that we can obtain comparable performance only by performing different early stopping for GT vs PosteL, why not do that?
>
>
> **Concerning Q3**, my concern is addressed. However, I just want to clarify that my question was meant to be, "Why does GPR-GNN or BernNet appear to not benefit from label smoothing **as much**." It would be useful to add the explanation that the authors provided in their response to the paper as well.
>
> A minor final point regarding Tables 1 and 5 of the paper: it is not self-evident which datasets are homophilic and which are heterophilic. This can be communicated more directly.
>
> [1] Zhou, Kaixiong, et al. "Adaptive label smoothing to regularize large-scale graph training." SDM 2023.

---

> > ### Author Response · Authors · 2024-08-13
> >
> > Thank you for your efforts in making constructive discussions.
> >
> > > Regarding Q1, the explanation provided is the type of information expected to be included in the paper. For the final revision of the paper, it is highly recommended that similarities be included and that the existence of [1] be acknowledged.
> >
> > We completely agree. We will revise the current text to clarify that our work is not the first to suggest label smoothing for node classification. In the revised version, we will cite [1] and [2] and explain the similarities and differences between their approaches and ours.
> >
> > > Likewise, it is important to highlight the key differences between each step in as much detail as possible. ALS's label propagation is similar to posterior label smoothing in this work, with the key difference being that the prior in the former is the hard labels while in the latter is the global statistics, which appears to be one of the elements that accommodate both homophilic and heterophilic graphs. A similar comparison is expected for the "label refinement" and "smooth pacing" steps from ALS. It would be great if the authors also provided such an explanation in their answer to this inquiry, which could further assist reviewers in their final opinion.
> >
> > Thank you for highlighting the key differences between existing work and ours.
> >
> > Both our method and ALS propose approaches for incorporating label relationships in label smoothing. In our method, the neighborhood label distribution captures these relationships directly from global statistics, whereas ALS uses label refinement, which involves training a separate function during the training process.
> >
> > Next, ALS's smooth pacing is a method that gradually increases the proportion of smoothed labels compared to GT labels. This approach could be directly adapted to our method to adjust the parameter $\alpha$.
> >
> > > Also, in Table 1 of the attached PDF, it would be great to highlight which datasets are homophilic and heterophilic.
> >
> > > A minor final point regarding Tables 1 and 5 of the paper: it is not self-evident which datasets are homophilic and which are heterophilic. This can be communicated more directly.
> >
> > Thank you for your feedback. The five datasets on the left are homophilic, while the five on the right are heterophilic. We will highlight this in the revised version.
> >
> > > Concerning the response to W3 & W4, while the current experiment conducted under section 4.3 is a valid experiment, it addresses the question of the efficiency of the model in learning. The title of the section, however, sets the expectation for the reader to see the performance of the model against different label sparsity ratios.
> > Here are some possibilities:
> > a) varying node label sparsity across the graph but with a fixed training set size.
> > b) varying node label sparsity across train and test sets. (so the sparsity is different per each set)
> > c) fixed label sparsity across the graph, but varying training set size while test set size is fixed.
> > The authors provide results for "case c", while "case a" seems to assess the robustness of the method against different sparsities because it addresses sparsity more directly, while "case c" includes the confounding effect of both train size and label sparsity.
> > My recommendation is to rephrase the current section title, and if authors intend to report robustness against sparsity, "case a" is a better option. If they decide not to include such an experiment in their work, it is then suggested to make necessary clarifications in the title and body of the section corresponding to this experiment, and they may mention this as a future work.
> >
> > We apologize for any confusion caused. We want to emphasize that in our setting, all of the training data is labeled. We use the term "sparse label" to refer to situations where the proportion of training data is small, resulting in a sparse labeled neighborhood. However, we understand that this terminology might confuse readers, so we will clarify the term and its explanation in the revised version.

---

> > > ### Author Response · Authors · 2024-08-13
> > >
> > > > Concerning Q2, First, a follow-up question: Loss curves for GT in Figure 2 from the paper and Figure 2 from the attached PDF do not match, e.g., GT labels in 1000 epochs get minimized up to ~0.4 loss, while in the new attached PDF for the same number of epochs, it gets minimized up to ~0.8. Similarly, for the PosteL, e.g., the difference between validation and test is more pronounced based on Figure 2 of the paper than the figure in the attachment. I was expecting them to be the same. Why are they different?
> > >
> > > The experimental settings for these figures differ. In Figure 2 of the paper, we conducted experiments using the same learning and model hyperparameters to compare the patterns of the loss curves. In contrast, Figure 2 of the attached PDF shows experiments conducted with the best hyperparameters for each baseline. Additionally, the loss curve in Figure 2 of the attached PDF represents the mean loss across 10 different splits, whereas the loss curve in Figure 2 of the paper shows the loss from a single split. Therefore, the loss curves may differ.
> > >
> > >
> > > > Second, I rephrase my question as I feel the main point is still not fully addressed: According to the original Figure 2 from the paper, early stopping at epoch ~50 provides a similar performance to Postel labels. That holds true for all datasets in Figure 6 and for some datasets in Figure 7. So, the question is, given that we can obtain comparable performance only by performing different early stopping for GT vs PosteL, why not do that?
> > >
> > > We believe there may have been a miscommunication regarding early stopping in our paper. We evaluate test performance based on the model parameters that achieve the lowest validation loss, so the performance of GT is evaluated around epoch 50. The 200 epochs mentioned refer to the patience set for early stopping, which is configured to prevent excessively long training times.
> > >
> > > It is an interesting observation that the loss for GT is similar to or even lower than that for PosteL. We found that even when the validaiton loss with the soft labels is greater than that of the GT labels, the accuracy of the PosteL is better than the model trained on GT labels. Due to the characteristics of the cross entropy loss, two different prediction results with the same label after argmax can have very different level of losses, e.g., for binary prediction, $\arg\max([0,1])=\arg\max([0.4,0.6])=1$. A similar phenomenon is observed in Figure 1 of [3], where some cases with label smoothing show higher loss than the baseline, yet their accuracy is greater. We want to emphasize that, despite this, our method often demonstrates lower test loss compared to GT, and the trend of the loss curve indicates a reduction in overfitting.
> > >
> > > > Concerning Q3, my concern is addressed. However, I just want to clarify that my question was meant to be, "Why does GPR-GNN or BernNet appear to not benefit from label smoothing **as much**." It would be useful to add the explanation that the authors provided in their response to the paper as well.
> > >
> > > Thank you for further explaining your question. We understand the point and will address it in the revised version.
> > >
> > > [1] Wang, Yiwei, et al. "Structure-aware label smoothing for graph neural networks." arXiv preprint (2021).
> > > [2] Zhou, Kaixiong, et al. "Adaptive label smoothing to regularize large-scale graph training." SDM 2023.
> > > [3] Xu, Yi, et al. "Towards understanding label smoothing." arXiv preprint (2020).

---

### Author Rebuttal · Authors · 2024-08-06

### **General response**

We sincerely appreciate the effort all reviewers dedicated to the review process. We are also grateful for the constructive feedback and have carefully considered all the comments we received. There are two questions that most reviewers asked. We address these questions in the general response. **Please refer to the attached [PDF] file at the bottom of the general response for the results of additional experiments.**

**1. Related work on label smoothing for node classification**

We recognize that [1,2] adopt similar approaches to improve GNNs by smoothing labels using the structural information of graph data. [1,2] are based on the assumption that the labels of connected nodes should be similar (homophily assumption). While this assumption may be beneficial for homophilic graphs, we believe that label smoothing based on the homophily assumption may be detrimental to heterophilic graphs. In contrast, PosteL does not rely on the homophily assumption. Instead, PosteL is based on the assumption that nodes with different labels should have distinct neighborhood label distributions. In Section 4.2 **Influence of Neighborhood Label Distribution**, we observe that this assumption holds for both homophilic and heterophilic graph datasets. We will add the similarities and distinctions of [1,2] in a future revision.

We compared the performance of PosteL against [1,2] with additional experiments. Since both [1] and [2] do not provide code, we have implemented these methods ourselves. We will make the code publicly available.

Table 1 in the attached [PDF] file shows the performance of the additional baselines. SALS refers to [1], and ALS refers to [2]. We conducted a hyperparameter search within the hyperparameter space used for the other methods (+KD, +LS, +PosteL). For the hyperparameters introduced in [1] and [2], we used the hyperparameter spaces specified in the original papers.

Our method outperforms SALS [1] and ALS [2] on both homophilic and heterophilic datasets. Specifically, our method demonstrates superior performance compared to SALS across all experimental settings and outperforms ALS in 62 out of 70 settings. Especially, *we observe a significant performance gap on heterophilic datasets*, which aligns with our assumption that label smoothing methods relying on the homophilic assumption should harm training for heterophilic datasets. We will add the performance and analysis of [1] and [2] in a future revision.

We also recognize that label propagation methods [3, 4] may appear similar to PosteL since they also mentioned *'label smoothing'*, but the use of *'label smoothing'* is quite different from ours. In Section 4.3 of [3] and Equation 8 of [4], label smoothing is proposed as a *post-processing* method of predicted labels. Specifically, they smoothed the predicted label through the neighborhood structure and used the smoothed labels as the final prediction of the model. They did not use the smoothed labels to further train the model. Therefore, their method can be considered as a post-processing method of the predicted label that can be independently applied to our method. Due to the difference, we do not compare the performance of these methods with PosteL. However, to avoid confusion, we will clarify this distinction in a revised version.

**2. Sensitivity analysis of the hyperparameters $\alpha$ and $\beta$**

We select $\alpha$ and $\beta$ based on validation performance. The values yielding the highest validation performance are selected. Figure 1 in the attached [PDF] file shows the performance with varying values of $\alpha$ and $\beta$ on GCN. The blue line indicates the performance with varying $\alpha$, and the green line shows the performance with varying $\beta$. The red dotted line represents the performance with the GT label.

We found three key takeaways from the results. First, regardless of the values of $\alpha$ and $\beta$, the performance consistently outperforms the case using GT labels, indicating that PosteL is insensitive to $\alpha$ and $\beta$. Second, $\alpha$ values greater than 0.8 may harm training, suggesting the necessity of interpolating GT labels. Lastly, searching within the ranges 0.5≤$\alpha$≤0.8 and $\beta$≤0.4 is sufficient. We intend to append this analysis in the revised version.


[1] Wang, Yiwei, et al. "Structure-aware label smoothing for graph neural networks." arXiv preprint (2021).
[2] Zhou, Kaixiong, et al. "Adaptive label smoothing to regularize large-scale graph training." SDM 2023.
[3] Zhang, Wentao, et al. "Node dependent local smoothing for scalable graph learning." NeurIPS 2021.
[4] Xie, Tian, Rajgopal Kannan, and C-C. Jay Kuo. "Label efficient regularization and propagation for graph node classification." IEEE TPAMI 2023.

[PDF]: https://openreview.net/attachment?id=wvWDdogE8H&name=pdf

---

### Decision · Program_Chairs · 2024-09-25

**Decision:**

Reject

**Comment:**

The reviewers have raised concerns about the limited novelty of the contributions, particularly due to similarities to Adaptive Label Smoothing (ALS) [1] and Structure-Aware Label Smoothing (SALS) [2]. While SALS is a preprint, it also raises questions about novelty. Although the authors clarified some differences, some concerns remained. The lack of theoretical analysis weakens the contribution further, and the empirical improvements are limited, with some lacking statistical significance.

Given these issues, the paper requires major revisions, and I recommend not accepting it in its current form.

[1] Zhou, Kaixiong, et al. “Adaptive Label Smoothing to Regularize Large-Scale Graph Training.” SDM 2023.

[2] Wang, Yiwei, et al. “Structure-Aware Label Smoothing for Graph Neural Networks.” arXiv preprint (2021).